# MULTI-AGENT PERFORMATIVE PREDICTION: FROM GLOBAL STABILITY AND OPTIMALITY TO CHAOS

## ABSTRACT

The recent framework of performative prediction (Perdomo et al., 2020) is aimed at capturing settings where predictions influence the target/outcome they want to predict. In this paper, we introduce a natural multi-agent version of this framework, where multiple decision makers try to predict the same outcome. We showcase that such competition can result in interesting phenomena by proving the possibility of phase transitions from stability to instability and eventually chaos. Specifically, we present settings of multi-agent performative prediction where under sufficient conditions their dynamics lead to global stability and optimality. In the opposite direction, when the agents are not sufficiently cautious in their learning/updates rates, we show that instability and in fact formal chaos is possible. We complement our theoretical predictions with simulations showcasing the predictive power of our results.

## 1 INTRODUCTION

Performative prediction (Perdomo et al., 2020) is a recently introduced framework that focuses on a natural but largely unexplored element of supervised learning. In many practical cases the predictive model can affect the very outcome that it is trying to predict. For example, predictions about which part of a road network will have high congestion trigger responses from the drivers which affect the resulting traffic realization leading to a shift of the target distribution. In such settings, (Perdomo et al., 2020) explored conditions for the existence and approximate optimality of stable equilibria of such processes.

One possible way to interpret the performative prediction setting is a single agent "game", where the predictive agent is playing a game against himself. An agent chooses the parameters of his model as his actions but the predictive accuracy/cost of the model depends on his own past actions. Fixed points of this process do not allow for profitable deviations. Once cast in this light, it becomes self-evident that the restriction to a single predictive agent/model is arguably only the first step in capturing more general phenomena where there is a closed loop between predictive models and their environment. This motivates our central question: *What is the interplay between stability, optimality in cases where multiple predictive models operate in parallel to each other? Can the competition between multiple models lead to novel phenomena such as phase transitions from stability to instability and chaos?*

A natural setting to consider for example is market competition, where multiple hedge funds are trying to simultaneously predict future prices, volatility of financial instruments. Of course as they act upon their predictions they also move the prices of these commodities in a highly correlated way. As more agents enter the market and the competition becomes increasingly fierce, is it possible that at some point the market flips from quickly discovering accurate stable predictions reflecting the underlying market fundamentals to self-induced unpredictability and chaos? When it comes to performative prediction can too many cooks spoil the soup?

**Our model** Standard supervised learning consists of three components: a set of predictive models, a loss function, and a data distribution. The learner (agent) observes samples of the distribution, and then decides a predictive model. When predictions are performative, the agent's decision of a predictive model influences the data distribution. Thus, instead of a fixed data distribution, a predictive prediction also has a *distribution map*: a mapping from the agent's predictive models to

data distributions. We further propose *multi-agent performative prediction* which model the influence of multiple agents' decisions on the data distribution, and ask whether these influence leads to convergent or chaotic systems.

To understand the framework of multi-agent performative prediction, we study a natural regression problem with multi-agent location-scale distribution maps (definition 2.2) where the agents' influence is linear in the agents' models. Specifically, the data consist of features and outcomes. For each agent $i \in [n]$, his influence on the distribution of outcome is his model weighted by a scalar $\lambda_i > 0$. When $n = 1$, our multi-agent location-scale distribution map is a special case of location-scale family in Miller et al. (2021). In the market example, each hedge fund company tries to predict the price (outcome) based on macroeconomic data or other information (features). These influence parameters $\lambda_1, \ldots, \lambda_n$ can be seen as each hedge fund's capital that can influence the distribution of the future price.

Finally, we consider the agents do not know the dependency between their model and the data distribution, and they can only improve their predictive models myopically and iteratively. In this paper, the agents myopically use a reinforcement learning algorithm, exponentiated gradient for linear regression (Kivinen & Warmuth, 1997), to improve their predictive models in rounds. We study the long-term behavior, and ask if the system converge to the performative stable and optimal point, or behave chaotically.

**Our results** We first study basic properties of our multi-agent preformative prediction with multi-agent location-scale distribution map. We show 1) the existence of performative stable point in our setting (proposition 3.1) and 2) the performative stability and performative optimality are equivalent (proposition 3.2). This equivalence allows us to focus on the dynamical behavior of the system.

In section 4, we introduce learning dynamics to the multi-agent performative prediction, and study their long-term behavior. We provide a threshold result which depends on the learning rates of exponentiated gradient descent and the collective influence. Theorem 4.1 shows the dynamics of exponential gradient descent converge to the performative stable and optimal point when the learning rate is small enough. Our convergence result in theorem 4.1 holds when the feature space is multi-dimensional, and every learning agent can use different learning rates starting at arbitrarily interior states. Our exact convergence result also works in the single-agent performative prediction setting. Contrarily, previous convergent results in Perdomo et al. (2020); Miller et al. (2021); Mendler-Dünner et al. (2020) only show that their dynamics converge to a small neighborhood of the performative stable point.

On the other hand, section 4.2 shows the dynamics can have chaotic behavior if the collective influence $L_n$ is large enough. Specifically, theorem 4.6 shows that even when the feature space is in $\mathbb{R}^2$ these systems provably exhibit Li-Yorke chaos, when the collective influence $\sum_i \lambda_i$ is large enough. This implies that there exists an uncountable "scrambled" set so that given any two initial conditions in the set, the liminf of the distance between these two dynamics is zero, but the limsup of their distance is positive. (definition 2.4) The chaotic result in theorem 4.6 also holds for the original single-agent performative prediction so long as the agent's influence is large enough, and, thus, complements previous performative prediction works on convergence behavior, which primarily consider that the agent's influence on the data distribution is sufficiently small. Moreover, no matter how small the agents' learning rates, theorem 4.6 shows that chaos is inevitable in some performative predictions settings when the number of agents exceeds a carrying capacity. After that the system becomes totally unpredictable with small perturbations exploding exponentially fast. [1]

Finally, section 5 provides numerical examples of our dynamics and show convergent and chaotic behavior. Additionally, through simulation, we demonstrate that our convergent and chaotic results also hold when the agents can only access noisy estimation of the gradient and conduct stochastic exponentiated gradient descent.

**Related work** Data distribution shift is not a new topic in ML, but earlier works focused primarily on exogenous changes to the data generating distribution. Performativity is machine learning was introduced by Perdomo et al. (2020). The original work and several follow-ups study the discrepancy between performative stability and performative optimality and prove approximated conver-

---

[1] We highlight our revision in blue color.

gence of learning dynamics, e.g., stochastic gradient descent, or iterative empirical risk minimization. (Mendler-Dünner et al., 2020; Drusvyatskiy & Xiao, 2020; Izzo et al., 2021) Performativity of prediction is also related to several applications: strategic classification (Hardt et al., 2016), retraining Bartlett (1992); Kuh et al. (1990).

Inspired by the instability of training algorithms in ML applications such as Generative Adversarial Networks (GANs), there has been a lot of recent interest in understanding conditions (particularly in multi-agent systems) where learning behavior may be non-equilibrating/unstable (Cheung & Tao, 2020; Balduzzi et al., 2020; Flokas et al., 2020; Andrade et al., 2021; Letcher, 2021; Giannou et al., 2021). The (in)stability and performance of exponentiated gradient descent in particular (also referred to as Multiplicative Weights Updates) and other closely related dynamics has attracted a lot of attention (Cohen et al., 2017; Bailey & Piliouras, 2018; Cheung, 2018; Panageas et al., 2019; Cheung & Piliouras, 2020; Vadori et al., 2021). The technique of Li-Yorke chaos has recently found applications across several different domains such as routing games, Cournot games and blockchain protocols (Palaiopanos et al., 2017; Chotibut et al., 2020; Bielawski et al., 2021; Cheung et al., 2021; Leonardos et al., 2021). To our knowledge, this is the first time where such formal chaotic results are established in settings related to performative prediction and supervised learning more generally.

Another line of related works is learning in games. Specifically, our dynamics can be seen as special cases multiplicative weight update (Hedge algorithms) on congestion games. Previous works on Hedge algorithms only show exact convergence when the learning rate is decreasing Kleinberg et al. (2009); Cohen et al. (2017), and, to our best knowledge, our results are the first that shows exact convergence of Hedge algorithms with small constant learning rates. Our results also complement the exact convergence result of the linear variant of multiplicative weight update by Palaiopanos et al. (2017).

## 2 PRELIMINARY

### 2.1 MULTI-AGENT PERFORMATIVE PREDICTION

A *multi-agent performative prediction* comprises $n$ agents deploying their predictive models $f_{\theta^1}, \ldots, f_{\theta^n}$ with parameters $\boldsymbol{\theta}^1, \boldsymbol{\theta}^2, \ldots, \boldsymbol{\theta}^n \in \Theta$ that collectively influence the future data distribution. We formalize such dependency via a *distribution map* $\mathcal{D}(\cdot)$ which outputs a distribution on the data, $\mathcal{D}(\vec{\boldsymbol{\theta}})$, given a *models profile* $\vec{\boldsymbol{\theta}} := (\boldsymbol{\theta}^1, \ldots, \boldsymbol{\theta}^n) \in \Theta^n$. A loss function $\ell(\boldsymbol{z}, \boldsymbol{\theta}')$ measures the loss of a model $f_{\boldsymbol{\theta}'}$ on a data point $\boldsymbol{z} \in \mathcal{Z}$, and the expected loss on a distribution $\mathcal{D}$ is $\mathbb{E}_{\mathbf{z} \sim \mathcal{D}}[\ell(\mathbf{z}, \boldsymbol{\theta}')]$. For performative prediction, we further define the decoupled performative loss on a distribution mapping $\mathcal{D}$ as

$$\ell(\vec{\boldsymbol{\theta}}, \boldsymbol{\theta}') := \mathbb{E}_{\mathbf{z} \sim \mathcal{D}(\vec{\boldsymbol{\theta}})} \ell(\mathbf{z}, \boldsymbol{\theta}')$$

where $\boldsymbol{\theta}' \in \Theta$ denotes a *predictive model*, while $\vec{\boldsymbol{\theta}} \in \Theta^n$ denotes a *deployed model profile*.

Given the set of model $\Theta$, the loss function $\ell$, and the distribution mapping $\mathcal{D}$, each agent in a multi-agent performative prediction $(\Theta, \ell, \mathcal{D})$ pursues minimal loss on the distribution that they collectively induce. We consider two solution concepts *performative optimality* and *performative stability* which generalizes the original ones in Perdomo et al. (2020).

**Definition 2.1.** Given $(\Theta, \ell, \mathcal{D})$, a models profile $\vec{\boldsymbol{\theta}}^* \in \Theta^n$ is *performatively optimal* if the total loss is minimized,

$$\vec{\boldsymbol{\theta}}^* \in \operatorname*{arg\,min}_{\vec{\boldsymbol{\theta}} \in \Theta^n} \sum_i \ell(\vec{\boldsymbol{\theta}}, \boldsymbol{\theta}^i).$$

Another desirable property of a model profile is that, given all agents deploy their models, their models are also simultaneously optimal for distribution that their model induces. Formally, $\vec{\boldsymbol{\theta}}^*$ is *performatively stable* if for all $i \in [n]$

$$(\boldsymbol{\theta}^*)^i \in \operatorname*{arg\,min}_{\boldsymbol{\theta}^i \in \Theta} \ell(\vec{\boldsymbol{\theta}}^*, \boldsymbol{\theta}^i).$$

The performative optimality does not implies the performative stable point. For performative optimal point, the variable for minimization $\vec{\boldsymbol{\theta}}$ affects both the first and the second argument, but only affect the second one for performative stable point. Now we introduce our model in this paper.

**Location-scale distribution map** In this paper, we study the family of multi-agent location-scale map for regression problem where a data point consists of $d$-dimensional feature and scalar outcome, $\boldsymbol{z} = (\boldsymbol{x}, y)$. These are natural classes of distribution maps in which performative effects enter through an additive or multiplicative factor that is linear in $\vec{\boldsymbol{\theta}}$.

**Definition 2.2.** Given $d \in \mathbb{N}$ and $d \geq 2$, and $\Theta^n \subseteq \mathbb{R}^d$, a distribution map $\mathcal{D} : \Theta^n \to \mathbb{R}^d \times \mathbb{R}$ is a *multi-agent location-scale distribution map on $n$ parties* if there exists a static distribution $\mathcal{D}_X$ on $\mathbb{R}^{d+1}$, $\boldsymbol{\theta}^0 \in \mathbb{R}^d$, and $n$ linear functions $\Lambda_1, \ldots, \Lambda_n$ from $\mathbb{R}^d$ to $\mathbb{R}^d$ so that the distribution of $(\mathbf{x}, \mathbf{y}) \sim \mathcal{D}(\vec{\boldsymbol{\theta}})$ has the following form: The feature is $\boldsymbol{x} \in \mathbb{R}^d$ and noise $x_0 \in \mathbb{R}$ is jointly sampled from $\mathcal{D}_X$. Given feature $\boldsymbol{x} \in \mathbb{R}^d$ and noise $x_0 \in \mathbb{R}$, and the the outcome is

$$y = \left\langle \boldsymbol{\theta}^0 - \sum_{i=1}^n \Lambda_i(\boldsymbol{\theta}^i), \boldsymbol{x} \right\rangle + x_0.$$

In this paper, we consider the scaling maps $\Lambda_i(\boldsymbol{\theta}) = \lambda_i \boldsymbol{\theta}$ for all $\boldsymbol{\theta} \in \Theta$ with scalar $\lambda_i > 0$ for all $i \in [d]$. We call $\boldsymbol{\lambda} := (\lambda_1, \ldots, \lambda_n)$ the influence parameters, and $L_n := \sum_{i=1}^n \lambda_i$ collective influence. Furthermore, we let $\boldsymbol{A} := \mathbb{E}\left[\mathbf{x}\mathbf{x}^\top\right] \in \mathbb{R}^{d \times d}$ be the covariance matrix of the feature, and $\boldsymbol{b} := \boldsymbol{A}\boldsymbol{\theta}^0 + \mathbb{E}[x_0\mathbf{x}] \in \mathbb{R}^d$. We will specify the multi agent location-scale distribution map with parameters $n, d, \boldsymbol{\lambda}, \boldsymbol{A}$, and $\boldsymbol{b}$.

When $n = 1$, our multi-agent location-scale distribution map is a special case of location-scale family in Miller et al. (2021) where the model $\boldsymbol{\theta}$ may both the outcome $y$ as well as the feature $\boldsymbol{x}$.

**Predictive Models and Loss Function** We consider *linear predictive model with constraint* where $f_{\boldsymbol{\theta}'}(\boldsymbol{x}) = \langle \boldsymbol{\theta}', \boldsymbol{x} \rangle$, and the collection of parameter is the $d$-simplex, $\Theta = \{\boldsymbol{\theta} : \sum_{k=1}^d \theta_k = 1, \theta_k \geq 0\}$. We use *mean squared error* to measure a predictive model $\boldsymbol{\theta}' \in \Theta$ on a distribution map with a deployed model profile $\vec{\boldsymbol{\theta}} \in \Theta^n$, $\ell(\vec{\boldsymbol{\theta}}, \boldsymbol{\theta}') = \mathbb{E}_{(\mathbf{x}, \mathbf{y}) \sim \mathcal{D}(\vec{\boldsymbol{\theta}})}[(\mathbf{y} - f_{\boldsymbol{\theta}'}(\mathbf{x}))^2] = \mathbb{E}_{(\mathbf{x}, \mathbf{y}) \sim \mathcal{D}(\vec{\boldsymbol{\theta}})}[(\mathbf{y} - \boldsymbol{\theta}' \cdot \mathbf{x})^2]$.

Given a deployed model profile $\vec{\boldsymbol{\theta}}$ and a predictive model $\boldsymbol{\theta}'$, the gradient of the decoupled loss is $\boldsymbol{g}(\vec{\boldsymbol{\theta}}, \boldsymbol{\theta}') := \nabla_{\boldsymbol{\theta}'} \ell(\vec{\boldsymbol{\theta}}, \boldsymbol{\theta}')$, and if $\mathcal{D}$ is a location-scale distribution map, $\boldsymbol{g}(\vec{\boldsymbol{\theta}}, \boldsymbol{\theta}') = \mathbb{E}_{(\mathbf{x}, \mathbf{y}) \sim \mathcal{D}(\vec{\boldsymbol{\theta}})}[2(\boldsymbol{\theta}' \cdot \mathbf{x} - \mathbf{y})\mathbf{x}]$. Furthermore, with $\boldsymbol{A}$ and $\boldsymbol{b}$ defined in definition 2.2, the gradient can be written as

$$\boldsymbol{g}(\vec{\boldsymbol{\theta}}, \boldsymbol{\theta}') = \nabla_{\boldsymbol{\theta}'} \ell(\vec{\boldsymbol{\theta}}, \boldsymbol{\theta}') = 2\boldsymbol{A}\left(\boldsymbol{\theta}' + \sum_i \lambda_i \boldsymbol{\theta}^i\right) - 2\boldsymbol{b}. \tag{1}$$

Additionally, given a deployed model profile $\vec{\boldsymbol{\theta}}$ and a predictive model profile $\vec{\boldsymbol{\theta}}'$, we define the gradient of agent $i$'s decoupled loss as $\boldsymbol{g}^i(\vec{\boldsymbol{\theta}}, \vec{\boldsymbol{\theta}}') := \boldsymbol{g}(\vec{\boldsymbol{\theta}}, (\boldsymbol{\theta}')^i)$, and $\boldsymbol{g}^i(\vec{\boldsymbol{\theta}}) := \boldsymbol{g}^i(\vec{\boldsymbol{\theta}}, \vec{\boldsymbol{\theta}})$ when the deployed model profile is identical to the predictive model profile. We denote the gradient of agent $i$'s average loss as $\bar{g}^i(\vec{\boldsymbol{\theta}}) := \sum_l \theta_l^i g_l^i(\vec{\boldsymbol{\theta}})$ for all $\vec{\boldsymbol{\theta}} \in \Theta^n$. Finally, we define $\vec{\boldsymbol{\xi}}(\vec{\boldsymbol{\theta}}) = (\boldsymbol{\xi}^1, \ldots, \boldsymbol{\xi}^n)$ with $\boldsymbol{\xi}^i \in \mathbb{R}^d$ so that for all $i \in [n]$ and $k \in [d]$ $\xi_k^i(\vec{\boldsymbol{\theta}}) := \theta_k^i(\sum_l \theta_l^i g_l^i - g_k^i)$. For brevity, we omit $\vec{\boldsymbol{\theta}}$ and define $\vec{\boldsymbol{g}} := \bar{\boldsymbol{g}}(\vec{\boldsymbol{\theta}})$ and $\vec{\boldsymbol{\xi}} := \vec{\boldsymbol{\xi}}(\vec{\boldsymbol{\theta}})$ when there is no ambiguity.

## 2.2 Learning Dynamics

We consider the agents myopically use a reinforcement learning algorithm *exponentiated gradient for linear regression* to improve their predictive models in rounds. We first define the original single agent's exponentiated gradient for linear regression on a sequence of data points here and will state our dynamics on multi agent performative prediction in section 4.

**Definition 2.3** (Kivinen & Warmuth (1997)). Given a learning rate $\eta > 0$, an initial parameter $\boldsymbol{\theta}_0 \in \Theta$ and a sequential of data points $(\boldsymbol{x}_t, y_t)_{t \geq 1}$, the *exponentiated gradient descent for linear regression* iteratively updates the model as follows: At round $t + 1$, with previous parameter $\boldsymbol{\theta}_t \in \Theta$ the exponentiated gradient algorithm updates it to $\boldsymbol{\theta}_{t+1}$ so that

$$\theta_{t+1,k} = \frac{\theta_{t,k} \exp(-2\eta(\boldsymbol{\theta}_t \cdot \boldsymbol{x}_t - y_t)x_{t,k})}{\sum_l \theta_{t,l} \exp(-2\eta(\boldsymbol{\theta}_t \cdot \boldsymbol{x}_t - y_t)x_{t,l})} \text{ for all } k \in [d].$$

Note that each exponent $2(\boldsymbol{\theta}_t \cdot \boldsymbol{x}_t - y_t)x_{t,k}$ is the $k$-th coordinate of the gradient of squared error $\ell((\boldsymbol{x}_t, y_t), \boldsymbol{\theta}_t)$ at $\boldsymbol{\theta}_t$ which yields the name of exponentiated gradient descent.

### 2.3 DYNAMICAL SYSTEM

**Definition 2.4.** Let $(\mathcal{X}, f)$ be a dynamical system where $\mathcal{X}$ is a metric space and $f$ is a mapping on $\mathcal{X}$. We say $(x, x') \in \mathcal{X} \times \mathcal{X}$ is a Li-Yorke pair if

$$\liminf_t dist(f^t(x), f^t(x')) = 0 \text{ and } \limsup_t dist(f^t(x), f^t(x') > 0.$$

$(\mathcal{X}, f)$ is Li-Yorke chaotic if there is an uncountable set $S \subset \mathcal{X}$ (scrambled set) such that any pair of points $x \neq x' \in S$ is Li-Yorke pair.

## 3 MULTI-AGENT PERFORMATIVE LEARNING: STABILITY AND OPTIMALITY

Having introduced multi-agent performative prediction, we show some basic property of our location-scale distribution map with mean squared error as a warm-up. Using the first order condition and a potential function argument, we show 1) the existence of performative stable point in our model (proposition 3.1) and 2) the performative stability and performative optimality are equivalent (proposition 3.2). The proofs of both propositions are in appendix A.

We first show the existence and uniqueness of performative stable point through a potential function argument. The first part is due to the first order condition of convex optimization. The second part also use the first order condition, and $\frac{\partial}{\partial \theta_k^i} \Phi(\vec{\theta}) = \lambda_i g_k^i(\vec{\theta})$ for all $i \in [n]$ and $k \in [d]$.

**Proposition 3.1** (existence). *Given the family of linear predictive models with constrains $\Theta$, mean squared error $\ell$ and multi-agent location-scale distribution map with parameters $n, d, \boldsymbol{\lambda}, \boldsymbol{A}, \boldsymbol{b}$ in definition 2.2, $\vec{\theta}^*$ on $(\Theta, \ell, \mathcal{D})$ is performative stable if and only if the gradient (defined in eq. (1)) satisfies $g_k^i(\vec{\theta}^*) \leq g_l^i(\vec{\theta}^*)$ for all $i \in [n]$ and $k \in [d]$ with $\theta_k^i > 0$.*

*Furthermore, if $\boldsymbol{A}$ is positive definite, there exists a unique performative stable point $\vec{\theta}^*$, and $\vec{\theta}^*$ is also a global minimizer of the following convex function for all $\vec{\theta} \in \Theta^n$*

$$\Phi(\vec{\theta}) := (\sum_i \lambda_i \boldsymbol{\theta}^i)^\top \boldsymbol{A}(\sum_i \lambda_i \boldsymbol{\theta}^i) + \sum_i \lambda_i (\boldsymbol{\theta}^i)^\top \boldsymbol{A}\boldsymbol{\theta}^i - 2\boldsymbol{b}^\top \sum_i \lambda_i \boldsymbol{\theta}^i. \tag{2}$$

While model profile is performative stable does not necessary mean the loss is minimized, the following proposition shows that optimality and stability are equivalent in our setting.

**Proposition 3.2.** *Given $(\Theta, \ell, \mathcal{D})$ defined in proposition 3.1, $\vec{\theta}^*$ is performative stable if and only if $\vec{\theta}^*$ is performatively optimal defined in definition 2.1.*

## 4 MULTI-AGENT PERFORMATIVE LEARNING: CONVERGENCE AND CHAOS

In section 3, we study the stability and optimality of location-scale distribution map with mean squared error. Now we ask when each agent myopically improves his predictive model through reinforcement learning but their predictions are performative, what is the long term behavior of the system? Can the system converges to performative stable and optimal point, or behave chaotically?

We provide a threshold result depending on the learning rate and the collective influence $L_n$. Section 4.1 shows the dynamics converge to the performative stable and optimal point when when the learning rate is small enough. On the other hand, section 4.2 shows the dynamics can have chaotic behavior if the collective influence $L_n$ is large enough.

Now, we define our dynamics $(\vec{\theta}_t)_{t \geq 0}$ formally. At each round, each agent accesses the data distribution which influenced by their previous model, and updates his model through exponentiated gradient descent. Specifically, given an initial model profile $\vec{\theta}_0 \in \Theta^n$ and a learning rate profile $\boldsymbol{\eta} := (\eta_1, \ldots, \eta_n)$, each agent $i$ applies the exponentiated gradient descent with initial parameter $\boldsymbol{\theta}_0^i$ and learning rate $\eta_i > 0$: At round $t + 1 > 1$, each agent $i$ use $\mathcal{D}(\vec{\theta}_t)$ and estimates the gradient of the (expected) loss, $\boldsymbol{g}^i(\vec{\theta}_t)$ defined in eq. (1), and updates his model according to definition 2.3,

$$\theta_{t+1,k}^i = \frac{\theta_{t,k}^i \exp(-\eta_i g_k^i(\vec{\theta}_t))}{\sum_{l=1}^d \theta_{t,l}^i \exp(-\eta_i g_l^i(\vec{\theta}_t))} \text{ for all } t \geq 0, i \in [n], \text{ and } k \in [d]. \tag{3}$$

We will use superscript to denote agent, $i \in [n]$, and subscript for time, $t \geq 0$, and index of feature, $k, l \in [d]$. Recall that the gradient of agent $i$'s average loss is $\bar{g}^i(\vec{\boldsymbol{\theta}}) := \sum_l \theta_l^i g_l^i(\vec{\boldsymbol{\theta}})$, and eq. (3) can be rewritten as $\frac{\theta_k^i \exp(-\eta_i g_k^i(\vec{\boldsymbol{\theta}}_t))}{\sum_l \theta_l^i \exp(-\eta_i g_l^i(\vec{\boldsymbol{\theta}}_t))} = \frac{\theta_k^i \exp(\eta_i(\bar{g}^i(\vec{\boldsymbol{\theta}}_t) - g_k^i(\vec{\boldsymbol{\theta}}_t)))}{\sum_l \theta_l^i \exp(\eta_i(\bar{g}^i(\vec{\boldsymbol{\theta}}_t) - g_l^i(\vec{\boldsymbol{\theta}}_t)))}$. Finally, given $\vec{\boldsymbol{\theta}}^*$ and $i \in [d]$, we define the support of $\vec{\boldsymbol{\theta}}^*$ as $S_i \subseteq [d] = \{k : (\theta^*)_k^i > 0\}$, and $\bar{S}_i := [d] \setminus S_i$. Then $\vec{\boldsymbol{\theta}}^*$ is an *equilibrium (or fixed point)* of eq. (3) if

$$g_k^i(\vec{\boldsymbol{\theta}}^*) = \bar{g}^i(\vec{\boldsymbol{\theta}}^*), \text{ for all } i \in [n], k \in S_i. \tag{4}$$

We say a fixed point of eq. (3) is *isolated* if there exists an open set of it so that no other fixed point is in the set. Additionally, the performative stable condition in proposition 3.1 is equivalent to

$$g_k^i(\vec{\boldsymbol{\theta}}^*) = \bar{g}^i(\vec{\boldsymbol{\theta}}^*) \text{ and } g_l^i(\vec{\boldsymbol{\theta}}^*) \geq \bar{g}^i(\vec{\boldsymbol{\theta}}^*), \text{ for all } i \in [n], k \in S_i, l \in \bar{S}_i. \tag{5}$$

Therefore, the set of fixed points of eq. (3) contains the performative stable point.

## 4.1 CONVERGING WITH SMALL LEARNING RATE

In this section, we show when the learning rate of each agent is small enough the the dynamics in eq. (3) converge to performative stable point. Specifically, if the parameter of multi-agent performative learning $(n, d, \boldsymbol{A}, \boldsymbol{\lambda})$ is fixed, the dynamics in eq. (3) converge to performative stable point when $\boldsymbol{\eta}$ is "small" enough.

By eq. (5), $\vec{\boldsymbol{\theta}}^*$ is a performative stable if the gradient in the support is no less than the average gradient. We call a performative stable point $\vec{\boldsymbol{\theta}}^*$ *proper* if the gradient of non-support coordinate is greater than the average gradient: for all $i \in [n]$ $l \in \bar{S}_i$, $g_l^i(\vec{\boldsymbol{\theta}}^*) > \bar{g}^i(\vec{\boldsymbol{\theta}}^*)$. The below theorem shows if the performative stable point is proper and equilibria satisfying eq. (4) are isolated, eq. (3) converges when $\max_i \eta_i$ is small enough and the ratio of $\eta_i/\eta_j$ is bounded for all $i$ and $j$.

**Theorem 4.1** (Convergence). *Consider a constant $R_\eta > 0$, and a multi-agent performative learning setting with parameter $n, d, \boldsymbol{A}, \boldsymbol{b}, \boldsymbol{\lambda}$ such that $\boldsymbol{A}$ is positive definite, the performative stable point $\vec{\boldsymbol{\theta}}^*$ is proper and its equilibria (defined in eq. (4)) are isolated. There exists $\eta_* > 0$ so that dynamic in eq. (3) with learning rate profile $\boldsymbol{\eta}$ converges to the performative stable point, $\lim_{t \to \infty} \vec{\boldsymbol{\theta}}_t = \vec{\boldsymbol{\theta}}^*$, if the initial state is an interior point and $\boldsymbol{\eta}$ satisfies $\max_i \eta_i \leq \eta_*$ and $\max_i \eta_i / \min_i \eta_i \leq R_\eta$.*

Informally, when the learning rate $\boldsymbol{\eta}$ are small, $\vec{\boldsymbol{\theta}}_t$ in eq. (3) can be approximated by a solution of an ordinary differential equation, $\vec{\boldsymbol{\vartheta}}(t\|\boldsymbol{\eta}\|_1)$, where the initial condition is $\vec{\boldsymbol{\vartheta}}(0) = \vec{\boldsymbol{\theta}}_0$ and

$$\frac{d}{dt} \vartheta_k^i(t) = \frac{\eta_i}{\|\boldsymbol{\eta}\|_1} \vartheta_k^i(t)(\bar{g}^i(\vec{\boldsymbol{\vartheta}}(t)) - g_k^i(\vec{\boldsymbol{\vartheta}}(t))) \tag{6}$$

for all $t \geq 0, i \in [n]$, and $k \in [d]$. We formalize this intuition in the proof of lemma 4.4. Note that the set of fixed points of eq. (6) is identical to eq. (3) and satisfies eq. (4).

The proof of theorem 4.1 has two parts. First we show the continuous approximation eq. (6) converges to the performative stable point. Then we study the relationship between eq. (3) and eq. (6), and prove the eq. (3) also converges to a performative stable point.

**From Potential Function to Convergence of Equation (6)** In this section, theorem 4.2 shows the dynamics of eq. (6) converge to performative stable point which will be useful to show the convergence of eq. (3).

**Theorem 4.2.** *If all points satisfying eq. (4) are isolated and $\boldsymbol{A}$ is positive definite, and $\vec{\boldsymbol{\vartheta}}(0)$ is in the interior of $\Theta^n$, the limit $\lim_{t \to \infty} \vec{\boldsymbol{\vartheta}}(t) = \vec{\boldsymbol{\theta}}^*$ is the performative stable point.*

Theorem 4.2 can be seen as the continuous version of theorem 4.1. Compared to theorem 4.1, theorem 4.2 does not require that the performative stable point is proper, but theorem 4.2 also implicitly requires the ratio of learning rate between any two agents is bounded.

To prove theorem 4.2. we prove lemma 4.3 which shows that $\Phi$ in eq. (2) is a potential function for eq. (6) so that time derivative of $\Phi$ is decreasing for all non-fixed points. Then, in the proof of theorem 4.2, we further prove that the dynamics in eq. (6) converge to performative stable points when the initial condition $\vec{\boldsymbol{\vartheta}}(0)$ is an interior point. The proof is similar to a proof in Kleinberg et al. (2009), and is presented in appendix B.

**Lemma 4.3.** *Given a solution of eq. (6), the time derivative of $\Phi$ in eq. (2) is 0 at fixed points of eq. (6), and negative at all other points. Furthermore,*

$$\frac{d}{dt}\Phi(\vec{\boldsymbol{\vartheta}}(t)) \le \frac{-1}{2\sum_i \eta_i \sum_i \lambda_i \eta_i} \left(\sum_{i,k} \lambda_i \eta_i |\vartheta_k^i(\sum_l \vartheta_l^i g_l^i - g_k^i)|\right)^2 \le \frac{-\min_i \lambda_i \eta_i}{2\sum_i \eta_i \sum_i \lambda_i \eta_i} \|\vec{\boldsymbol{\xi}}(\vec{\boldsymbol{\vartheta}})\|_1^2.$$

**From Approximation to Convergence of Equation (3)** Given the convergence of eq. (6), we show the dynamics of eq. (3) also converges to a performative stable point. The argument has two stages. We first show given any neighborhood of performative stable points $D$, eq. (3) will hits $D$ and stay in $D$ in finite amount of steps in lemma 4.4. Then lemma 4.5 shows that eq. (3) converges to a performative stable point which completes the proof of theorem 4.1.

**Lemma 4.4.** *Given any open set $D \in \Theta^n$ that contains the performative stable point, there exists $\eta_*$ small enough so that for all $\vec{\boldsymbol{\theta}}_0 \in \Theta^n$, there exists $\tau$ so that $\vec{\boldsymbol{\theta}}_t \in D$ for all $t \ge \tau$.*

The proof is based on standard approximation result of numerical solution to ordinary differential equations, and is presented in appendix C.

**Lemma 4.5.** *Let $\vec{\boldsymbol{\theta}}^*$ is the performative stable point and $D$ is a small enough neighborhood of $\vec{\boldsymbol{\theta}}^*$. Besides the condition in theorem 4.1, if the initial condition of eq. (3), $\vec{\boldsymbol{\theta}}_0$, is in $D$ and $\vec{\boldsymbol{\theta}}_t \in D$ for all $t \ge 0$, the limit of eq. (3) is the performative stable point $\lim_{t \to \infty} \vec{\boldsymbol{\theta}}_t = \vec{\boldsymbol{\theta}}^*$.*

The proof uses the fact that the right-hand side of eq. (3) decreases as the dynamic converges to the fixed point. Therefore, even though the learning rate profile is fixed, the error between eq. (3) and eq. (6) vanishes as they converge to the fixed point. We present the proof in appendix D.

## 4.2 Chaos with Constant Learning Rate

Now we want to ask the converse question in section 4.1. Do the dynamics in eq. (3) still converge, if the learning rate is fixed, as the number of agent increases? Alternatively, do the dynamics converge when the agents have overwhelming influence on the data distribution?

Theorem 4.6 shows that the dynamics is Li-York chaotic when $L_n = \sum_i \lambda_i$ is large even with $d = 2$. Note that large $L_n$ may be due to a fixed number of agent which have overwhelming influence, or the number of agents is large. The later case implies for any small $\eta$, no matter how cautious the agents are, as number of agent increases the chaos inevitably occurs.

**Theorem 4.6.** *Given a multi-party induced location scale family distribution with $\boldsymbol{A}$, $\boldsymbol{b}$, $d = 2$, and a common learning rate $\eta > 0$, if all $n$ agents use the exponentiated gradient with a common learning rate $\eta$ in eq. (3) and $\boldsymbol{A}$ is diagonally dominant, there exists a carrying capacity $L^*$ such that if $L_n = \sum_{i=1}^n \lambda_i \ge L^*$ the dynamics $(\vec{\boldsymbol{\theta}}_t)_{t\in\mathbb{N}}$ is Li-Yorke chaotic and thus has periodic orbits of all possible periods.*

To prove theorem 4.6, we show there exists an uncountable scrambled set defined in definition 2.4. First we consider all agents start from the same initial model, and show the dynamics eq. (3) can be captured by the following function,

$$f_{u,v}(x) = \frac{x}{x + (1-x)\exp(u(x-v))} \quad \forall x \in [0,1] \tag{7}$$

with proper choice of $u \in \mathbb{R}$ and $v \in \mathbb{R}$. Note that $v$ is an equilibrium, and $u$ controls steepness. First, when all agents start from the same initial model, $\boldsymbol{\theta}_0^i = \boldsymbol{\theta}_0$ for all $i \in [n]$, and use the same learning rate $\eta$, their models are identical for all $t \ge 0$. For all $t$ there exists $\boldsymbol{\theta}_t$ so that $\boldsymbol{\theta}_t^i = \boldsymbol{\theta}_t$ for all $i$. Additionally, since $d = 2$, we can use a single parameter to encode a linear predictive model with constraints. Given $\boldsymbol{\theta} \in \Theta = \Delta_2$, we define $p(\boldsymbol{\theta}) = \theta_1 \in [0,1]$ and omit the input $\boldsymbol{\theta}$ when it is clear. Thus, we can rewrite the dynamics as $p_t = p(\boldsymbol{\theta}_t)$ which are one-dimensional dynamics on $[0,1]$, and $p_{t+1} = \frac{p_t}{p_t + (1-p_t)e^{\eta(g_1^i(\vec{\boldsymbol{\theta}}_t) - g_2^i(\vec{\boldsymbol{\theta}}_t))}}$. We set

$$\alpha(L) = 2\eta(1+L)(A_{1,1} - A_{1,2} - A_{2,1} + A_{2,2}) \text{ and } \beta(L) = \frac{(1+L)(A_{2,2} - A_{2,1}) + (b_1 - b_2)}{(1+L)(A_{1,1} - A_{1,2} - A_{2,1} + A_{2,2})},$$

and write $\alpha_n = \alpha(L_n)$ and $\beta_n = \beta(L_n)$. By direct computation the dynamic of $p_t$ is

$$p_{t+1} = f_{\alpha_n, \beta_n}(p_t) \tag{8}$$

where $\alpha_n$ encodes the update pace and $\beta_n$ is an equilibrium of the dynamic. We further set $\beta_\infty := \frac{A_{2,2} - A_{2,1}}{A_{1,1} - A_{1,2} - A_{2,1} + A_{2,2}}$ and $\delta(L) := \beta(L) - \beta_\infty$ which converges to zero as $L \to \infty$. If $\boldsymbol{A}$ is diagonally dominant $\alpha(L)$ is positive and increasing for all $L \geq 0$. Additionally, because $A_{1,1} - A_{1,2}$ and $A_{2,2} - A_{2,1}$ are positive, we can permute the coordinate so that $\beta_\infty \in (0, 1/2]$. With Li et al. (1982), the lemma below implies the existence of period three points and the proof is in appendix E.

**Lemma 4.7.** *If $\beta_\infty \in (0, 1/2)$, there exists a constant $L^* > 0$ so that if $L \geq L^*$, there exists a trajectory $x_0, x_1, x_2$, and $x_3$ with $f_{\alpha(L), \beta(L)}(x_i) = x_{i+1}$ such that $x_3 < x_0 < x_1$.*

*Proof of theorem 4.6.* First by lemma 4.7, for all $L \geq L^*$, there exists a trajectory $x_0, x_1, x_2, x_3$ so that $x_3 < x_0 < x_1$. Additionally, existence of such trajectory implies the exists of period three points by Li et al. (1982). Finally, by the seminal work of Li & Yorke (1975), it implies that the map is Li-York chaotic. □

## 5 SIMULATION

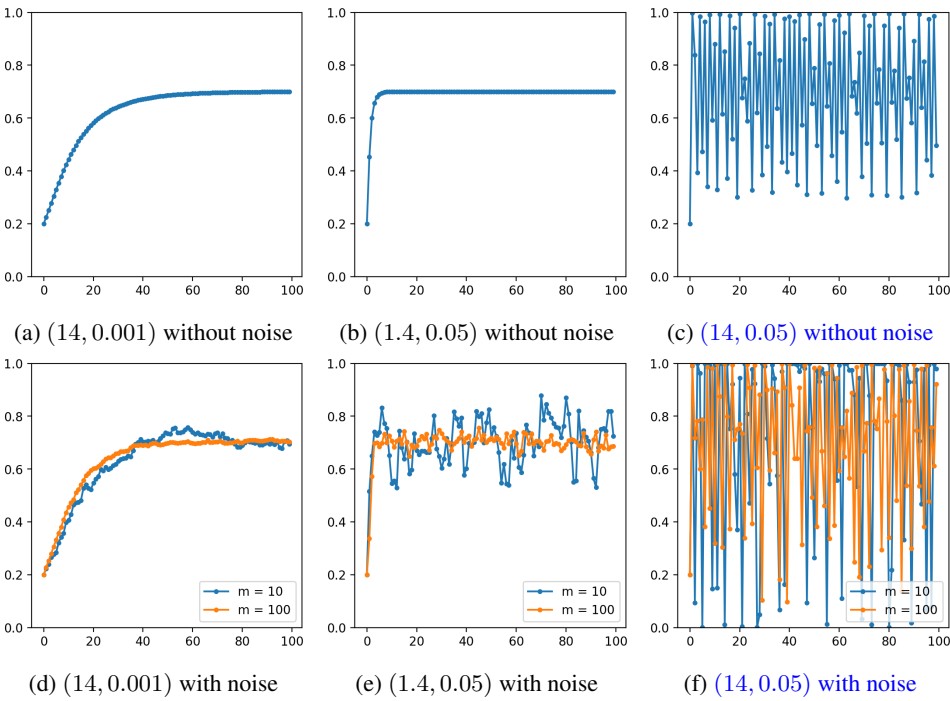

(a) $(14, 0.001)$ without noise   (b) $(1.4, 0.05)$ without noise   (c) $(14, 0.05)$ without noise

(d) $(14, 0.001)$ with noise   (e) $(1.4, 0.05)$ with noise   (f) $(14, 0.05)$ with noise

Figure 1: Here we plot the temporal behavior of $p_t$ starting at $p_0 = 0.2$ for 100 rounds under various $L$ and $\eta$, and noisy estimation of gradient of mean squared error with $m$ samples. The top row (figs. 1a to 1c) present different combination of $(L, \eta)$, and bottom row (figs. 1d to 1f) consider the gradient is estimated from $m = 10$ and $m = 100$ samples.

Now we simulate the one-dimensional dynamics in eq. (8) of one agent with different learning rate $\eta$ and collective influence $L$.

We first define a location-scale distribution map. Let the feature $x_1, x_2$ and noise $x_0$ are mutually independent Gaussian distribution with zero mean, and the variance are $\mathbb{E}[x_1^2] = 3$, $\mathbb{E}[x_2^2] = 7$, and $\mathbb{E}[x_0^2] = 1$. Finally, $\boldsymbol{\theta}^0 = \boldsymbol{0}$. Therefore,

$$\boldsymbol{A} = \mathbb{E}[\mathbf{x}\mathbf{x}^\top] = \begin{bmatrix} 3 & 0 \\ 0 & 7 \end{bmatrix}, \text{ and } \boldsymbol{b} = \boldsymbol{A}\boldsymbol{\theta}^0 + \mathbb{E}[x_0\mathbf{x}] = \begin{bmatrix} 0 \\ 0 \end{bmatrix}.$$

Given learning rate $\eta > 0$, and collective influence $L$, we have $\alpha(L) = 20\eta(1 + L)$, and the performative stable point $\beta(L) = 0.7$.

The top row of fig. 1 shows the temporal behavior of $p_t$ under different $(L, \eta)$. The bottom row in fig. 1 demonstrates our (convergent and chaotic) results are robust even when the value of gradient is noisy. Specifically, we consider at each round $t$, instead of $g_1^i(\vec{\theta}_t) - g_2^i(\vec{\theta}_t) = 2(1+L)(1, -1)\mathbf{A}\boldsymbol{\theta}_t - 2(1, -1)\mathbf{b}$, the agent replace $\mathbf{A}$ and $\mathbf{b}$ with empirical moments on $m$ samples. This process can be seen as a stochastic exponentiated gradient descent which uses a stochastic estimation of the gradient.

We can see chaotic behavior happens in fig. 1c with $L = 14$ and $\eta = 0.05$, and such behavior persists in fig. 1f when the evaluations of gradient are noisy. On the other hand, when the learning rate is small enough (figs. 1a and 1b) the dynamics converge to the performative stable point 0.7. Furthermore, in figs. 1d and 1e the dynamics converge even with noisy estimation of gradient.

Finally, fig. 2 shows the temporal behavior of the loss. Under the same setting as fig. 1c, fig. 2b shows not only the temporal behavior $p_t$ is chaotic, but also the mean squared loss. On the other hand, under the same setting as fig. 1a, fig. 2a shows convergent behavior of loss.

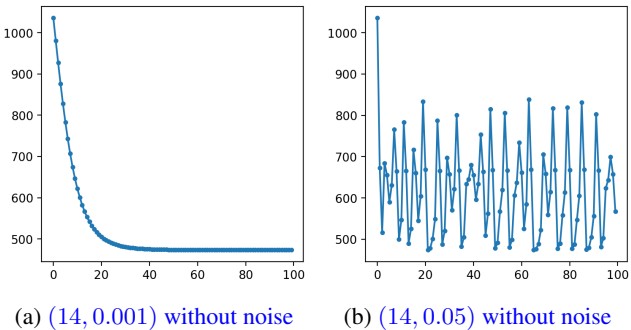

(a) $(14, 0.001)$ without noise     (b) $(14, 0.05)$ without noise

Figure 2: The temporal behavior of the total cost for 100 rounds under various $\eta$.

## 6   CONCLUSION

We introduce a framework of multi-agent performative prediction and investigate whether classical reinforcement learning algorithms can converge or behave chaotically depending on the collective influence of the agents model and learning rate. However, we view our example as only scratching the surface of the work of multi-agent performative predictions.

Our framework leads to several new theoretical problems. In particular, it would be interesting to understand whether this threshold is generic and if our results still hold on other reinforcement learning algorithms or other general multi-agent distribution maps. Our framework also provides a new viewpoint to several applications. One natural application is strategic classification. Hardt et al. (2016) In this context, our framework can be seen as strategic learners in strategic classification problem where both data and classifiers are strategic. However, the features also respond to the deployed models in conventional strategic classification, which is not captured in our multi-agent location-scale distribution map. It would be interesting to investigate our dynamics in the context of strategic prediction. Another application is pricing strategy/prediction, where multiple companies predict the demand function and set their prices.

Both our digital and real-world environments are increasingly under the influence of or ever more powerful and numerous ML systems. As these algorithms trigger actions that change the state of the system that produces their joint input data (e.g., AI-triggered ads shown to users affecting their behavior, or automated trading systems affecting stock prices, etc), they are effectively forming closed loop systems and can no longer be understood fully in isolation. As we show in this paper, even fairly simple and innocuous such settings can exhibit phase transitions going from optimal behavior to chaos. We believe such phenomena are worthy of a careful investigation and we hope that this paper sets out some useful building blocks by bringing together optimization/performance analysis with Lyapunov theory and chaos theory.

## ETHICS STATEMENT

Our work showcases the possibility of competing prediction algorithms leading to instability and chaos. Extra steps should be taken, whenever possible, so that related systems operate within the range of parameters leading to stability. We hope that our work helps both by identifying a potential threat for ML systems as well as providing some guidance on how to counter or minimize it.

## REPRODUCIBILITY

All of our theoretical results are stated fully, including previous definitions and results on which they are based. They are accompanied by proofs, either in the main paper or in the appendix. The setting of our simulation results are stated fully in the main paper.

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

## A    PROOFS AND DETAILS IN SECTION 3

*Proof of proposition 3.1.* First under the mean squared loss function, each agent $i$'s decoupled performative loss function $\ell(\vec{\boldsymbol{\theta}}', \boldsymbol{\theta}^i)$ is convex in $\boldsymbol{\theta}^i$. Thus, for linear predictive model, we can apply the KKT conditions on definition 2.1 so that a collection of predictive models $\vec{\boldsymbol{\theta}} \in \Theta^n$ is performatively stable if and only if for all $i \in [n]$, $k \in [d]$ with $\theta_k^i > 0$, $\frac{\partial}{\partial \theta_k^i}\ell(\vec{\boldsymbol{\theta}}', \boldsymbol{\theta}^i) \le \frac{\partial}{\partial \theta_l^i}\ell(\vec{\boldsymbol{\theta}}', \boldsymbol{\theta}^i)$ for all $l \in [d]$ where $\vec{\boldsymbol{\theta}}' = \vec{\boldsymbol{\theta}}$. Therefore, with eq. (1), $\vec{\boldsymbol{\theta}}^*$ is performative stable if

$$g_k^i(\vec{\boldsymbol{\theta}}^*) \le g_l^i(\vec{\boldsymbol{\theta}}^*) \tag{9}$$

holds for all $i \in [n]$ and $k \in [d]$ with $\theta_k^i > 0$.

With eq. (9), we now show that there exists a unique performative stable point $\vec{\boldsymbol{\theta}}^*$ by proving that 1) $\Phi$ in eq. (2) is strictly convex, and 2) $\vec{\boldsymbol{\theta}}^*$ is a minimizer of $\Phi$. First, to show $\Phi$ is strictly convex, it is sufficient to show the Hessian of $\Phi$ positive definite. Because $\Phi$ is a quadratic function on $\Theta^n$, the Hessian of $\Phi$ is a constant matrix in $\mathbb{R}^{nd \times nd}$ By the partial derivative of eq. (2), for all $i, j \in [n]$ and $k, l \in [d]$, we have $(\nabla^2 \Phi)_{ik,jl} = \frac{\partial^2}{\partial \theta_k^i \partial \theta_l^j}\Phi = 2\lambda_i\lambda_j A_{k,l}$ if $i \ne j$ and $(\nabla^2 \Phi)_{ik,il} = 2\lambda_i(\lambda_i + 1)A_{k,l}$. Let $\boldsymbol{L} \in \mathbb{R}^{n \times n}$ with $L_{ij} = 2\lambda_i\lambda_j$ if $i \ne j$ and $2\lambda_i(1 + \lambda_i)$ which is positive definite because $\lambda_i > 0$ for all $i \in [n]$. Then, the Hessian of $\Phi$ is the Kronecker product of $\boldsymbol{L}$ and $\boldsymbol{A}$,

$$\nabla^2 \Phi = \begin{pmatrix} 2\lambda_1(1 + \lambda_1)\boldsymbol{A} & \dots & 2\lambda_1\lambda_n\boldsymbol{A} \\ \vdots & \ddots & \vdots \\ 2\lambda_n\lambda_1\boldsymbol{A} & \dots & 2\lambda_n(1 + \lambda_n)\boldsymbol{A} \end{pmatrix} = \boldsymbol{L} \otimes \boldsymbol{A}.$$

Because $\boldsymbol{L}$ and $\boldsymbol{A}$ are both positive definite, the Hessian is also positive definite. Therefore, $\Phi$ is strictly convex, and there exist a unique minimum of $\Phi$ in the compact set $\Theta^n$. Now we show $\vec{\boldsymbol{\theta}}^*$ is performative stable if and only if $\vec{\boldsymbol{\theta}}^*$ is a minimizer of $\Phi$. By the first order condition and the partial derivative of eq. (2), the minimum of eq. (2) at $\vec{\boldsymbol{\theta}}^*$ if and only if $g_k^i(\vec{\boldsymbol{\theta}}^*) \le g_l^i(\vec{\boldsymbol{\theta}}^*)$ for all $i \in [n]$ $k, l \in [d]$ with $(\theta^*)_k^i > 0$. Therefore, $\vec{\boldsymbol{\theta}}^*$ is the minimum of $\Phi$ if and only if $\vec{\boldsymbol{\theta}}^*$ is a performative stable point. $\square$

*Proof of proposition 3.2.* Given a profile of models $\vec{\boldsymbol{\theta}}$, the total cost is

$$\sum_i \ell(\vec{\boldsymbol{\theta}}, (\boldsymbol{\theta})^i) = \sum_i \mathbb{E}_{(\mathbf{x},\mathbf{y}) \sim \mathcal{D}(\vec{\boldsymbol{\theta}})}[(\mathbf{y} - \boldsymbol{\theta}^i \cdot \mathbf{x})^2]$$

$$= \sum_i \mathbb{E}\left[\left(\left\langle \mathbf{x}, \boldsymbol{\theta}^0 - \sum_j \lambda_j\boldsymbol{\theta}^j - \boldsymbol{\theta}^i \right\rangle + x_0\right)^2\right]$$

which is a convex function on $\vec{\boldsymbol{\theta}} \in \Theta^n$. Additionally,

$$\frac{\partial}{\partial \theta_k^i}\sum_i \ell(\vec{\boldsymbol{\theta}}, \boldsymbol{\theta}^i) = (1 + \lambda_i)g_k^i(\vec{\boldsymbol{\theta}}, \vec{\boldsymbol{\theta}}) + \sum_{i \in [n]: i \ne i} \lambda_i g_k^i(\vec{\boldsymbol{\theta}}, \vec{\boldsymbol{\theta}}) = (1 + n\lambda_i)g_k^i(\vec{\boldsymbol{\theta}}, \vec{\boldsymbol{\theta}})$$

which is the gradient of decoupled performative loss scaled by $(1 + n\lambda_i)$. Thus, we can apply the KKT conditions and the minimum happens if and only if eq. (9) holds. $\square$

## B    PROOF AND DETAILS FOR THEOREM 4.2

*Proof of lemma 4.3.* By chain rule,

$$\frac{d}{dt}\Phi(\vec{\boldsymbol{\vartheta}}(t)) = \sum_{i \in [n], k \in [d]} \frac{\partial}{\partial \theta_k^i}(\vec{\boldsymbol{\vartheta}}(t)) \cdot \frac{d}{dt}\vec{\boldsymbol{\vartheta}}(t), \tag{10}$$

so the time derivative of the potential function is zero if the dynamics is at a fixed point of eq. (6).

Now we compute a closed form of the time derivative and show the derivative is zero only if the dynamics is at a fixed point of eq. (6). Here we omit the input $t$ to simplify the notation.

$$\frac{d}{dt}\Phi(\vec{\vartheta}(t)) = \sum_{i,k}\frac{\partial}{\partial\theta_k^i}(\vec{\vartheta}(t))\cdot\frac{\eta_i}{\|\boldsymbol{\eta}\|_1}\vartheta_k^i(t)(\sum_l\vartheta_l^i(t)g_l^i(\vec{\vartheta}(t)) - g_k^i(\vec{\vartheta}(t))) \quad \text{(by eqs. (6) and (10))}$$

$$=\frac{1}{\|\boldsymbol{\eta}\|_1}\sum_i\lambda_i\eta_i\left(\sum_k\left(g_k^i\right)\cdot\vartheta_k^i\left(\sum_l\vartheta_l^ig_l^i - g_k^i\right)\right)$$
$$\text{(by the partial derivative of eq. (2))}$$

$$=\frac{1}{2\|\boldsymbol{\eta}\|_1}\sum_i\lambda_i\eta_i\sum_{k,l\in[d]}\vartheta_k^i\vartheta_l^i(2g_k^ig_l^i - 2(g_k^i)^2) \qquad \text{(because } \sum_l\vartheta_l^i = 1\text{)}$$

$$=\frac{-1}{2\|\boldsymbol{\eta}\|_1}\sum_i\lambda_i\eta_i\sum_{k,l\in[d]}\vartheta_k^i\vartheta_l^i(g_k^i - g_l^i)^2$$

Now we bound the time derivative.

$$-2\|\boldsymbol{\eta}\|_1\left(\sum_i\lambda_i\eta_i\right)\frac{d}{dt}\Phi(\vec{\vartheta}(t))$$

$$=\left(\sum_{i\in[n],k,l\in[d]}\lambda_i\eta_i\vartheta_k^i\vartheta_l^i\right)\left(\sum_{i\in[n],k,l\in[d]}\lambda_i\eta_i\vartheta_k^i\vartheta_l^i(g_k^i - g_l^i)^2\right) \qquad (\sum_k\vartheta_k^i = \sum_l\vartheta_l^i = 1)$$

$$\geq\left(\sum_{i\in[n],k,l\in[d]}\lambda_i\eta_i\vartheta_k^i\vartheta_l^i|g_k^i - g_l^i|\right)^2. \qquad \text{(by Cauchy inequality)}$$

$$\geq\left(\sum_{i\in[n],k\in[d]}\lambda_i\eta_i\left|\vartheta_k^i\sum_{l\in[d]}\vartheta_l^i(g_k^i - g_l^i)\right|\right)^2 \qquad \text{(by triangle inequality)}$$

Therefore, $\frac{d}{dt}\Phi(\vec{\vartheta}(t)) = 0$ only if $\vartheta_k^i\left(g_k^i - \sum_{l\in[d]}\vartheta_l^ig_l^i\right) = 0$ for all $i\in[n]$ and $k\in[d]$ which is a fixed point of eq. (6).

We can further simplify the bound by $\vec{\xi}$. Because $\sum_{i,k}\lambda_i\eta_i|\vartheta_k^i(\sum_l\vartheta_l^ig_l^i - g_k^i)| \geq (\min_i\lambda_i\eta_i)\sum_{i,k}|\xi_k^i(\vec{\vartheta})| = (\min_i\lambda_i\eta_i)\|\vec{\xi}(\vec{\vartheta})\|_1$, $\frac{d}{dt}\Phi(\vec{\vartheta}(t)) \leq \frac{-\min_i\lambda_i\eta_i}{2\sum_i\eta_i\sum_i\lambda_i\eta_i}\|\vec{\xi}(\vec{\vartheta})\|_1^2$. $\qquad\square$

*Proof of theorem 4.2.* When the fixed points are isolated satisfying eq. (4), by lemma 4.3, the dynamics in eq. (6) converges to a fixed point of eq. (6), $\vec{\theta}^*\in\Theta^n$. Because $\vec{\theta}^*$ is a fixed point, for all $i\in[n], k\in[d]$,

$$(\theta^*)_k^i\left(\bar{g}^i(\vec{\theta}^*) - g_k^i(\vec{\theta}^*)\right) = (\theta^*)_k^i\sum_l(\theta^*)_l^ig_l^i(\vec{\theta}^*) - g_k^i(\vec{\theta}^*) = 0$$

By proposition 3.1, the fixed point condition implies each coordinate of the gradient of loss in the support are identical, $g_k^i(\vec{\theta}^*) = \bar{g}^i(\vec{\theta}^*)$ for all $i$ and $k\in S_i$. Thus, if the fixed point $\vec{\theta}^*$ is not performative stable, there exists $\imath$ and $\kappa$ with $(\theta^*)_\kappa^\imath = 0$ so that $g_\kappa^\imath(\vec{\theta}^*) < \bar{g}^\imath(\vec{\theta}^*)$. Furthermore, $\exp(-\eta_\imath g_\kappa^\imath(\vec{\theta}^*)) > \sum_l(\theta^*)_l^\imath\exp(-\eta_\imath g_l^\imath(\vec{\theta}^*))$. We can pick a small enough $\epsilon > 0$ and define

$$U_\epsilon := \{\vec{\theta}: \exp(-\eta_\imath g_\kappa^\imath) > \sum_l\theta_l^\imath\exp(-\eta_\imath g_l^\imath) + \epsilon\} \tag{11}$$

which contains $\vec{\theta}^*$ and is an open set because $\vec{g}$ and the exponential function are continuous. Since $\vec{\vartheta}(t)$ converges to $\vec{\theta}^*$ as $t\to\infty$, there exists a time $t_\epsilon$ so that for all $t\geq t_\epsilon$, $\vec{\vartheta}(t)\in U_\epsilon$. However, if $\vec{\vartheta}(t)\in U_\epsilon$ and $\vec{\vartheta}(t)$ is an interior point, we get $\frac{d}{dt}\vartheta_k^\imath(t) > 0$ by eqs. (6) and (11). Therefore, $\vartheta_k^\imath(t)$ is positive and increasing for $t\geq t_\epsilon$. We reached a contradiction because $\vartheta_k^\imath(t)\to(\theta^*)_k^\imath = 0$. Therefore, $\vec{\theta}^*$ is a performative stable point. $\qquad\square$

## C    PROOF ANS DETAILS FOR LEMMA 4.4

To prove lemma 4.4, show the dynamic eq. (3) can be approximated by eq. (6) and the error vanishes as $\eta_*$ decreases. Thus, we can show the dynamic can hit an arbitrary neighborhood of the performative stable point. We further use $\Phi$ to show the dynamic will stay in the neighborhood. Below we state two ancillary lemmas to control the error of our approximation.

We define an error vector $\vec{e}(\vec{\theta}) \in \mathbb{R}^{n \times d}$ between eqs. (3) and (6) so that

$$\eta_i^2 e_k^i(\vec{\theta}) := \frac{\theta_k^i \exp(\eta_i(\bar{g}^i - g_k^i))}{\sum_l \theta_l^i \exp(\eta_i(\bar{g}^i - g_l^i))} - \theta_k^i - \eta_i \xi_k^i$$

We omit the input $\vec{\theta}$ of $\vec{e}$ and $\vec{e}_t := \vec{e}(\vec{\theta}_t)$ when there is no ambiguity. We show if the the error vector is small, $\Phi$ is decreasing in dynamics eq. (3).

**Claim C.1** (Approximated potential). *There exist $C_4$, so that for all $t$ and $\vec{\theta}_0 \in \Theta^n$, the difference of eq. (2) on eq. (3) satisfies*

$$\Phi(\vec{\theta}_{t+1}) - \Phi(\vec{\theta}_t) \leq \frac{-\min_i \lambda_i^2 \eta_i^2}{2 \sum_i \lambda_i \eta_i} \|\vec{\xi}_t\|_1^2 + \max_i \lambda_i \eta_i^2 \max_{i,k,\vec{\theta}} |g_k^i(\vec{\theta})| \cdot \|\vec{e}_t\|_1 + d^2 C_4 \left\|\vec{\theta}_{t+1} - \vec{\theta}_t\right\|_1^2.$$

**Claim C.2.** *If $\max_i \eta_i$ is small enough and satisfies eq. (12), there exists a constant $C$ so that*

$$|e_k^i(\vec{\theta})| \leq C \text{ for all } i \in [n], k \in [d], \text{ and } \vec{\theta} \in \Theta^n.$$

The proofs of these two claims are based on first order approximation.

*Proof of claim C.1.* By the partial derivative of eq. (2), we have $\frac{\partial^2}{\partial \theta_k^i \partial \theta_l^j} \Phi(\vec{\theta}) = 2\lambda_i \lambda_j A_{l,k}$ if $i \neq j$ and $\frac{\partial^2}{\partial \theta_k^i \partial \theta_l^i} \Phi(\vec{\theta}) = 2\lambda_i(1 + \lambda_i) A_{l,k}$. Thus, the second order partial derivatives of $\Phi$ can bounded by the twice of $C_4 := \max_i \lambda_i(\lambda_i + 1) \max_{k,l} A_{k,l}$. By Taylor's expansion, we have

$$\Phi(\vec{\theta}_{t+1}) - \Phi(\vec{\theta}_t)$$
$$\leq \nabla\Phi(\vec{\theta}_t) \cdot \left(\vec{\theta}_{t+1} - \vec{\theta}_t\right) + \frac{d^2 2C_4}{2} \left\|\vec{\theta}(t+1) - \vec{\theta}(t)\right\|_2^2$$
$$\leq \sum_{i \in [n], k \in [d]} \lambda_i g_k^i(\vec{\theta}_t)(\eta_i \xi_k^i(\vec{\theta}_t) + \eta_i^2 e_k^i(\vec{\theta}_t)) + d^2 C_4 \left\|\vec{\theta}(t+1) - \vec{\theta}(t)\right\|_1^2$$

$$\text{(by the partial derivative of eq. (2))}$$

$$= \sum_{i \in [n], k \in [d]} \lambda_i \eta_i g_{t,k}^i \xi_{t,k}^i + \lambda_i \eta_i^2 g_{t,k}^i e_{t,k}^i + d^2 C_4 \left\|\vec{\theta}_{t+1} - \vec{\theta}_t\right\|_1^2$$

$$\leq \frac{-1}{2\sum_i \lambda_i \eta_i} \left(\sum_{i,k} \lambda_i \eta_i |\xi_{t,k}^i|\right)^2 + \sum_{i \in [n], k \in [d]} \lambda_i \eta_i^2 g_{t,k}^i e_{t,k}^i + d^2 C_4 \left\|\vec{\theta}_{t+1} - \vec{\theta}_t\right\|_1^2 \text{ (by lemma 4.3)}$$

Therefore, we have

$$\Phi(\vec{\theta}_{t+1}) - \Phi(\vec{\theta}_t) \leq \frac{-\min_i \lambda_i^2 \eta_i^2}{2 \sum_i \lambda_i \eta_i} \|\vec{\xi}_t\|_1^2 + \max_i \lambda_i \eta_i^2 \max_{i,k,\vec{\theta}} |g_k^i(\vec{\theta})| \cdot \|\vec{e}_t\|_1 + d^2 C_4 \left\|\vec{\theta}_{t+1} - \vec{\theta}_t\right\|_1^2$$

which completes the proof.    □

*Proof of claim C.2.* By if $\eta_*$ is small enough and satisfies eq. (12), we have $\eta_i(\bar{g}^i - g_k^i) \leq 1$, and $\theta_k^i + \eta_i \theta_k^i(\bar{g}^i - g_k^i) \leq \theta_k^i e^{\eta_i(\bar{g}^i - g_k^i)} \leq \theta_k^i + \eta_i \theta_k^i(\bar{g}^i - g_k^i) + \frac{e}{2}\eta_i^2 \theta_k^i(\bar{g}^i - g_k^i)^2$ for all $i, k$ and $\vec{\theta} \in \Theta^n$. Additionally, with $\xi_k^i = \theta_k^i(\bar{g}^i - g_k^i)$, we can rewrite it as

$$\theta_k^i \leq \theta_k^i e^{\eta_i(\bar{g}^i - g_k^i)} - \eta_i \xi_k^i \leq \theta_k^i + \frac{e}{2}\eta_i^2 \theta_k^i(\bar{g}^i - g_k^i)^2.$$

Therefore, the error can be upper bounded as

$$\eta_i^2 e_k^i \leq \frac{\theta_k^i + \eta_i \xi_k^i + \frac{e}{2}\eta_i^2 \theta_k^i (\bar{g}^i - g_k^i)^2}{\sum_l \theta_l^i + \eta_i \xi_l^i} - (\theta_k^i + \eta_i \xi_k^i) = \frac{e}{2}\eta_i^2 \theta_k^i (\bar{g}^i - g_k^i)^2,$$

because $\sum_l \theta_l^i = 1$ and $\sum_l \xi_l^i = 0$. For lower bound, we have,

$$\eta_i^2 e_k^i \geq \frac{\theta_k^i + \eta_i \xi_k^i}{\sum_l \theta_l^i + \eta_i \xi_l^i + \frac{e}{2}\eta_i^2 \theta_l^i (\bar{g}^i - g_l^i)^2} - (\theta_k^i + \eta_i \xi_k^i) \geq -(\theta_k^i + \eta_i \xi_k^i)\left(\frac{e\eta_i^2}{2}\sum_l \theta_l^i(\bar{g}^i - g_l^i)^2\right).$$

If we set $C := e\max_{\vec{\theta}}\left|\sum_l \theta_l^i (\bar{g}^i - g_l^i)^2\right|$, because $|\eta_i \xi_k^i| \leq \eta_i 2\max_k |g_k^i| \leq 1$ by eq. (12) and $\theta_k^i \leq 1$, we have $\left|e_k^i(\vec{\theta})\right| \leq C$. $\qquad\square$

*Proof of lemma 4.4.* Because $D$ is open, we can set $D'$ so that $\vec{\theta}^* \in D'$, $D' \subset D$, and $\sup_{\vec{\theta}\in D'} \Phi(\vec{\theta}) < \frac{1}{2}\inf_{\vec{\theta}\notin D} \Phi(\vec{\theta})$. We will pick $\eta_*$ small enough so that $\vec{\theta}_{t+1} \in D$ for all $\vec{\theta}_t \in D' \subset D$. The proof has two parts: we first show the dynamics eq. (3) hits the set $D$. Then we prove eq. (3) stays in $D$ afterward.

By theorem 4.2, there exists $\tau$ so that $\vec{\vartheta}(\tau) \in D'$. Now by Gronwall's inequality we can set $\eta_*$ small enough so that $\vec{\theta}_{\tau/\|\eta\|_1} \in D$. Formally, given the dynamics in eq. (3), we define right-continuous step functions $\vec{\theta}(t) = \vec{\theta}_{\lfloor t \rfloor}$ and $\vec{e}(t) := \vec{e}(\vec{\theta}_{\lfloor t \rfloor})$ for all $t \geq 0$. Then the dynamics in eq. (3) can be written as

$$\theta_k^i(t) - \theta_k^i(0) = \sum_{s=0}^{t-1}(\theta_{s+1,k}^i - \theta_{s,k}^i) = \sum_{s=0}^{t-1}\left(\eta_i \xi_k^i(\vec{\theta}_s) + \eta_i^2 e_k^i(\vec{\theta}_s)\right) = \eta_i \int_0^t \xi_k^i(\vec{\theta}(s)) + \eta_i e_k^i(s)\,ds$$

On the other hand, the solution of eq. (6) can be written as

$$\vartheta_k^i(\eta_i t) - \vartheta_k^i(0) = \frac{\eta_i}{\|\eta\|_1}\int_0^{\frac{t}{\|\eta\|_1}} \xi_k^i(\vec{\vartheta}(s))\,ds = \eta_i \int_0^t \xi_k^i(\vec{\vartheta}(\|\eta\|_1 s))\,ds$$

Because $\xi_k^i$ are continuous and $\Theta^n$ is compact, there exists $L > 0$ so that $\xi_k^i$ is $L$-Lipschitz with respect to one norm for all $i$ and $k$. Thus, the difference between above equations is

$$\left|\theta_k^i(t) - \vartheta_k^i(\|\eta\|_1 t)\right| \leq \eta_i \int_0^t \left|\xi_k^i(\vec{\theta}(s)) - \xi_k^i(\vec{\vartheta}(\|\eta\|_1 s))\right|\,ds + \eta_i^2\int_0^t |e_k^i(s)|\,ds$$

$$\leq \eta_i L \int_0^t \left\|\vec{\theta}(s) - \vec{\vartheta}(\eta_i s)\right\|_1\,ds + \eta_i^2 Ct \qquad (L\text{-Lipschitz and claim C.2})$$

Therefore, by Gronwall's inequality and $\eta_i \leq \eta_*$, we have

$$\left\|\vec{\theta}(t) - \vec{\vartheta}(\|\eta\|_1 t)\right\|_1 \leq \eta_* ndL \int_0^t \left\|\vec{\theta}(s) - \vec{\vartheta}(\|\eta\|_1 s)\right\|_1\,ds + (\eta_*)^2 ndCt \leq (\eta_*)^2 ndCt e^{\eta_* ndLt}$$

Because $\vec{\vartheta}(\tau) \in D'$, we can pick $\eta_*$ small enough so that $\vec{\theta}(\tau/\|\eta\|_1) = \vec{\theta}_{\lfloor\tau/\|\eta\|_1\rfloor} \in D$ and $\Phi(\vec{\theta}_{\lfloor\tau/\|\eta\|_1\rfloor}) < \inf_{\vec{\theta}\notin D} \Phi(\vec{\theta})$.

Now we show the second part that $\vec{\theta}_t \in D$ for all $t \geq \lfloor\tau/\|\eta\|_1\rfloor$. First, with claim C.1, we will prove that the potential function is decreasing for all $\vec{\theta}_t \notin D'$ when $\eta_*$ small enough. We estimate three terms in claim C.1 separately. First, because $\vec{\theta}^* \in D'$ and $\Theta^n\backslash D'$ is compact, $\min_{\vec{\theta}\notin D'} \|\vec{\xi}(\vec{\theta})\|_1 > 0$ exists. Then $\frac{\min_i \lambda_i^2 \eta_i^2}{2\sum_i \lambda_i \eta_i}\|\vec{\xi}_t\|_1^2 = \Omega(\max_i \eta_i)$ By claim C.2, $\max_i \lambda_i \eta_i^2 \max_{i,k,\vec{\theta}} |g_k^i(\vec{\theta})| \cdot \|\vec{e}_t\|_1 = O(\max_i \eta_i^2)$. Finally, $d^2 C_4\left\|\vec{\theta}_{t+1} - \vec{\theta}_t\right\|_1^2 = O(\max_i \eta_i^2)$. Therefore, there exists $\eta_*$ small enough so that the potential function $\Phi(\vec{\theta}_{t+1}) - \Phi(\vec{\theta}_t) < 0$ for all $\vec{\theta}_t \notin D'$ and $\eta$. Therefore $\Phi(\vec{\theta}_t) < \min_{\vec{\theta}\notin D} \Phi(\vec{\theta})$ for all $t \geq \lfloor\tau/\|\eta\|_1\rfloor$ which completes the proof. $\qquad\square$

# D    PROOFS AND DETAILS FOR LEMMA 4.5

Lemma 4.3, shows the value of $\Phi$ is decreasing in eq. (6), and the decrease rate is lower bounded by the one norm of $\vec{\xi}$. Thus, if we can show the error between eq. (3) and eq. (6) is bounded by $\|\vec{\xi}\|_1$, we have the value of $\Phi$ is also decreasing on eq. (3). The main challenge is that because $\vec{\xi}(\vec{\theta}^*) = \mathbf{0}$, we need to control the error as $\vec{\xi}$ converges to zero but $\eta_*$ is fixed.

We first show three ancillary claims, claim D.1 to D.3. Claim D.1 show the vanishing components of $\vec{\theta}_t$ decrease rapidly once the dynamic is in $D$. Claim D.2 and D.3 show the supporting component also decrease once the vanishing components are small enough.

Given $(n, d, \boldsymbol{A}, \boldsymbol{\lambda})$, we define the following constants $C_1 := \frac{1}{4} \max_{i,l,\boldsymbol{\theta} \in \Theta^n} |g_l^i|$, $C_2 := \frac{2}{\min_{i,k}(\theta^*)_k^i}$, $C_3 := ed \max(C_1, C_2)$, and $C_4 := \max_i \lambda_i(\lambda_i + 1) \max_{k,l} A_{k,l}$. We require the maximum learning rate is bounded by $\eta_*$ which satisfies the following conditions.

$$\eta_* \max_{i,k,\vec{\theta} \in \Theta^n} |g_k^i(\vec{\theta})| \leq \frac{1}{2} \tag{12}$$

$$\eta_* n d C_3 \max_{\vec{\theta}} \|\vec{\xi}(\vec{\theta})\|_1 \leq 1 \tag{13}$$

Additionally, given the bound of learning rate ratio, $\frac{\max_i \eta_i}{\min_i \eta_i} \leq R_\eta$, we requires

$$R_\eta^2 \eta_*^3 < \frac{\min_i \lambda_i^2}{16 \sum_i \lambda_i} \min \left\{ \frac{4}{C_3 \max_i(\lambda_i) \max_{\vec{\theta}} \|\vec{g}\|_1}, \frac{1}{d^2 C_4} \right\} \tag{14}$$

Note that $\frac{\max \eta_i^3}{\min \eta_i^2}$ is less then the right hand side of eq. (14). On the other hand, by lemma 4.4, we can pick $D$ small enough so that the following conditions holds.

$$\frac{1}{2} \min_{i \in [n], k \in S_i} (\theta^*)_k^i \leq \min_{i \in [n], k \in S_i, \vec{\theta} \in D} \theta_k^i \tag{15}$$

We first show for all $i$ and $k \in \bar{S}_i$, $\theta_{t,k}^i$ is decreasing and converges to zero exponentially fast as $t$ increases. Because $\vec{\theta}^*$ is a proper performative stable, $\sum_l (\theta^*)_l^i e^{-\eta_i g_l^i(\vec{\theta}^*)} = e^{-\eta_i \bar{g}^i(\vec{\theta}^*)} > e^{-\eta_i g_k^i(\vec{\theta}^*)}$ for all $i$ and $k \in \bar{S}_i$. We can take $\eta_*$, $\epsilon_1$, and $D$ small enough so that for all $\vec{\theta} \in D$ and all learning rate profile $\boldsymbol{\eta}$ with $\max \eta_i \leq \eta_*$,

$$e^{-\eta_i g_k^i} < (1 - \epsilon_1) \sum_l \theta_l^i e^{-\eta_i g_l^i}, i \in [n], k \in \bar{S}_i \tag{16}$$

**Claim D.1** (Vanishing components). *Given $\eta_*$, $\epsilon_1$, and $D$ in eq. (16), if $\vec{\theta}_t \in D$ for all $t \geq 0$, for all $i \in [n]$ and $k \in S_i$, $\theta_k^i(t)$ is decreasing in $t$ and for all $t \geq 0$*

$$0 \leq \theta_{t,k}^i \leq \theta_{0,k}^i e^{-\epsilon_1 t}.$$

*Proof of claim D.1.* By eq. (16), for all $t \geq 0$, $\theta_{t+1,k}^i = \theta_{t,k}^i \frac{\exp(-\eta_i g_{t,k}^i)}{\sum_l \theta_{t,l}^i \exp(-\eta_i g_{t,l}^i)} \leq (1 - \epsilon_1)\theta_{t,k}^i$. Therefore, $\theta_{t,k}^i$ is decreasing, and $\theta_{t,k}^i \leq \theta_{0,k}^i e^{-\epsilon_1 t}$. $\qquad\square$

**Claim D.2.** *There exists a constant $C_3$ such that for all $\vec{\theta} \in \Theta^n$ with $\delta_1 > 0$ so that $\|\vec{\xi}(\vec{\theta})\|_1 \geq 2\sqrt{\delta_1}$, and $|\xi_k^i(\vec{\theta})| \leq \delta_1$ for all $i \in [n]$ and $k \in \bar{S}_i$, then*

$$|e_k^i(\vec{\theta})| \leq C_3 \|\vec{\xi}(\vec{\theta})\|_1^2, \tag{17}$$

*for all $i \in [n]$ and $k \in [d]$.*

Claim D.2 shows if the one norm of $\vec{\xi}$ is much bigger than the vanishing components, the error term $\vec{e}$ can be bounded by the $\|\vec{\xi}\|_1^2$. Moreover, claim D.1 ensures that the vanishing components decrease rapidly, so the condition of claim D.2 readily holds.

*Proof of claim D.2.* Given $\vec{\theta}$ and $\delta_1 > 0$ that satisfy the condition, we first show two inequalities to bound $\theta_k^i(\bar{g}^i - g_k^i)^2$ for supporting and vanishing component respectively. For a vanishing component $k \in \bar{S}_i$,

$$\theta_k^i(\bar{g}^i - g_k^i)^2 \leq \max_{i,l,\boldsymbol{\theta} \in \Theta^n} |g_l^i| \cdot |\theta_k^i(\bar{g}^i - g_k^i)| \leq 4C_1 \cdot \delta_1 \leq C_1 \|\vec{\xi}\|_1^2. \tag{18}$$

Then for a supporting component $k \in S_i$, with eq. (15) we have

$$\theta_k^i(\bar{g}^i - g_k^i)^2 \leq \frac{1}{\min_{i,l,\vec{\theta} \in D} \theta_l^i}(\theta_k^i(g^i - g_k^i))^2 \leq \frac{2}{\min_{i,k}(\theta^*)_k^i}|\xi_k^i|^2 \leq C_2\|\vec{\xi}\|_1^2. \tag{19}$$

Now we use above two inequalities to approximate eq. (3). For nominator, because $1 + x \leq \exp(x) \leq 1 + x + \frac{e}{2}x^2$ for all $x \leq 1$, $\theta_k^i \exp(\eta_i(\bar{g}^i - g_k^i)) \geq \theta_k^i + \eta_i\theta_k^i(\bar{g}^i - g_k^i) = \theta_k^i + \eta_i\xi_k^i$. On the other hand, because $\eta_i(\bar{g}^i - g_k^i) \leq 1$ by eq. (12), $\theta_k^i \exp(\eta_i(\bar{g}^i - g_k^i)) \leq \theta_k^i + \eta_i\xi_k^i + \frac{e}{2}\theta_k^i\eta_i^2(\bar{g}^i - g_k^i)^2$. By eqs. (18) and (19), we have

$$0 \leq \theta_k^i e^{\eta_i(\bar{g}^i - g_k^i)} - \theta_k^i - \eta_i\xi_k^i \leq \frac{e}{2}\max(C_1, C_2)\eta_i^2\|\vec{\xi}\|_1^2 \tag{20}$$

For denominator, we sum over eq. (20). Because $\sum_l \theta_l^i = 1$ and $\sum_l \theta_l^k(\bar{g}^i - g_l^i) = 0$, we have

$$0 \leq \sum_l \theta_l^i e^{\eta_i(\bar{g}^i - g_l^i)} - 1 \leq \frac{ed}{2}\max(C_1, C_2)\eta_i^2\|\vec{\xi}\|_1^2 \tag{21}$$

Given $i \in [n]$ and $k \in [d]$, we apply the above equation to eq. (3). For upper bounds, we have

$$\begin{aligned}
\eta_i^2 e_k^i(\vec{\theta}_t) &= \frac{\theta_k^i \exp(\eta_i(\bar{g}^i - g_k^i))}{\sum_l \theta_l^i \exp(\eta_i(\bar{g}^i - g_l^i))} - \theta_{t,k}^i - \eta_i\xi_{t,k}^i \\
&\leq \theta_{t,k}^i e^{-\eta_i g_k^i(\vec{\theta}_t)} - \theta_{t,k}^i - \eta_i\xi_{t,k}^i &\text{(by eq. (21))} \\
&\leq \frac{e\max(C_1, C_2)}{2}\eta_i^2\|\vec{\xi}\|_1^2 &\text{(by eq. (20))}
\end{aligned}$$

For lower bounds,

$$\begin{aligned}
\eta_i^2 e_k^i(\vec{\theta}_t) &= \frac{\theta_k^i \exp(\eta_i(\bar{g}^i - g_k^i))}{\sum_l \theta_l^i \exp(\eta_i(\bar{g}^i - g_l^i))} - \theta_{t,k}^i - \eta_i\xi_{t,k}^i \\
&\geq \frac{\theta_{t,k}^i + \eta_i\xi_{t,k}^i}{1 + \frac{ed\max(C_1,C_2)}{2}\eta_i^2\|\vec{\xi}\|_1^2} - \theta_{t,k}^i - \eta_i\xi_{t,k}^i &\text{(by eqs. (20) and (21))} \\
&\geq -(\theta_{t,k}^i + \eta_i\xi_{t,k}^i)\left(\frac{ed\max(C_1,C_2)}{2}\eta_i^2\|\vec{\xi}\|_1^2\right) &(1/(1+x) \geq 1 - x) \\
&\geq -ed\max(C_1, C_2)\eta_i^2\|\vec{\xi}\|_1^2 &(\theta_{t,k}^i \leq 1 \text{ and } \eta_i\xi_{t,k}^i \leq 1 \text{ by eq. (12)})
\end{aligned}$$

Therefore, with $C_3 := ed\max(C_1, C_2)$ we finish the proof. $\square$

**Claim D.3.** *There exists $\epsilon_2 > 0$ so that for all $\vec{\theta}_t \in \Theta^n$ and $\delta_1 > 0$ so that $\|\vec{\xi}(\vec{\theta}_t)\|_1 \geq 2\sqrt{\delta_1}$, and $|\xi_k^i(\vec{\theta}_t)| \leq \delta_1$ for all $i \in [n]$ and $k \in \bar{S}_i$, then $\Phi(\vec{\theta}_{t+1}) - \Phi(\vec{\theta}_t) \leq \epsilon_2\|\vec{\xi}(\vec{\theta}_t)\|_1^2$.*

*Proof for claim D.3.* By claim C.1,

$$\Phi(\vec{\theta}_{t+1}) - \Phi(\vec{\theta}_t) \leq \frac{-\min_i \lambda_i^2\eta_i^2}{2\sum_i \lambda_i\eta_i}\|\vec{\xi}_t\|_1^2 + \max_i \lambda_i\eta_i^2 \max_{i,k,\vec{\theta}}|g_k^i(\vec{\theta})| \cdot \|\vec{e}_t\|_1 + d^2C_4\left\|\vec{\theta}_{t+1} - \vec{\theta}_t\right\|_1^2.$$

We can bound the later two terms by claim D.2. For the second term, $\|\vec{e}_t\|_1 \leq ndC_3\|\vec{\xi}_t\|_1^2$. For the third term,

$$\begin{aligned}
\left\|\vec{\theta}_{t+1} - \vec{\theta}_t\right\|_1 &\leq \sum_{i,k}\left(\eta_i|\xi_{t,k}^i| + C_3\eta_i^2\|\vec{\xi}\|_1^2\right) \\
&\leq \max_i \eta_i\|\vec{\xi}\|_1 + ndC_3 \max_i \eta_i^2\|\vec{\xi}\|_1^2. \\
&\leq 2\max_i \eta_i\|\vec{\xi}\|_1 &(\text{ by eq. (13)})
\end{aligned}$$

With above inequalities, by eq. (14), we have $\frac{\min_i \lambda_i^2 \eta_i^2}{4 \sum_i \lambda_i \eta_i} > \max_i \lambda_i \eta_i^2 \max_{i,k,\vec{\theta}} |g_k^i(\vec{\theta})| n d C_3$ and $\frac{\min_i \lambda_i^2 \eta_i^2}{4 \sum_i \lambda_i \eta_i} > 4 d^2 C_4 \max_i \eta_i^2$. Therefore there exists a constant

$$\epsilon_2 := \frac{\min_i \lambda_i^2 \eta_i^2}{2 \sum_i \lambda_i \eta_i} - \max_i \lambda_i \eta_i^2 \max_{i,k,\vec{\theta}} |g_k^i(\vec{\theta})| n d C_3 - 4 d^2 C_4 \max_i \eta_i^2 > 0$$

so that $\Phi(\vec{\theta}_{t+1}) - \Phi(\vec{\theta}_t) < -\epsilon_2 \|\vec{\xi}_t\|_1^2$. $\qquad \square$

*Proof of lemma 4.5.* To prove $\lim_{t\to\infty} \vec{\theta}_t = \vec{\theta}^*$, there are two equivalent ways to measure the progress, besides $\|\vec{\theta}_t - \vec{\theta}^*\|_1$. First because $\vec{\theta}^*$ is an isolated fixed point of eq. (6), $\vec{\theta}_t$ converges to $\vec{\theta}^*$ if and only if $\lim_{t\to\infty} \vec{\xi}(\vec{\theta}_t) = \vec{0}$. On the other hand, because $\Phi$ is strictly convex and and $\vec{\theta}^*$ is the minimum point, $\vec{\theta}_t$ converges to $\vec{\theta}^*$ if and only if $\lim_{t\to\infty} \Phi(\vec{\theta}_t) = \min_{\vec{\theta}} \Phi(\vec{\theta})$. With these two equivalent conditions, given any $\epsilon > 0$, there exists $\delta > 0$ so that $\|\vec{\theta} - \vec{\theta}^*\|_1 < \epsilon$ when

$$\vec{\theta} \in V_\delta := \left\{ \vec{\theta} : \Phi(\vec{\theta}) \leq \max_{\vec{\theta}' : \|\vec{\xi}(\vec{\theta}')\|_1 \leq \delta} \Phi(\vec{\theta}') \right\}.$$

Therefore, it is sufficient for us to show for all $\delta > 0$, there is $t_\delta$ so that $\vec{\theta}_t \in V_\delta$ for all $t \geq t_\delta$. With technical claim D.1 to D.2, our proof has two parts. First we show that the dynamic hits $V_\delta$. Then, the dynamic stays in $V_\delta$.

For the first part, given $\delta > 0$ and $0 < C < 1$, by claim D.1 there exists $T_1$ such that each vanishing component $|\xi_{t,k}^i| \leq C^2 \delta^2 / 4$ for all $t \geq T_1$. Then by claim D.3, there exists $T_2 \geq T_1$ such that $\|\vec{\xi}_{T_2}\|_1 \leq C\delta$. Otherwise, the value of $\Phi$ decreases by a nonzero constant $\epsilon_2 \|\vec{\xi}_t\|_1^2 \geq C \epsilon_2 \delta^2 > 0$ at each round which contradicts that the minimum of $\Phi$ bounded.

For the second part, if $\|\vec{\xi}_t\|_1 \leq C\delta \leq \delta$ for all $t \geq T_2$, then we finish the proof. Otherwise, there exists $\tau \geq T_2$ so that $\|\vec{\xi}_\tau\|_1 \leq C\delta < \|\vec{\xi}_{\tau+1}\|_1$. Now we prove that $\Phi(\vec{\theta}_{\tau+1}) \leq \max_{\vec{\theta}' : \|\vec{\xi}(\vec{\theta}')\|_1 \leq \delta} \Phi(\vec{\theta}')$. Because the difference between $\vec{\theta}_{\tau+1}$ and $\vec{\theta}_\tau$ is $\|\vec{\theta}_{\tau+1} - \vec{\theta}_\tau\|_1 \leq \max_i \eta_i \|\vec{\xi}_\tau\|_1 + \max_i \eta_i^2 \|\vec{e}_\tau\|_1 \leq 2 \max_i \eta_i \|\vec{\xi}_\tau\|_1$ when $\eta_*$ is small enough, and $\vec{\xi}$ is a $L_\xi$-Lipschitz $\vec{\theta}$ for some constant $L_\xi$ in one norm, we have

$$C\delta < \|\vec{\xi}_{\tau+1}\|_1 \leq \|\vec{\xi}_\tau\|_1 + \|\vec{\xi}_{\tau+1} - \vec{\xi}_\tau\|_1 \leq \|\vec{\xi}_\tau\|_1 + 2 L_\xi \max_i \eta_i \|\vec{\xi}_\tau\|_1 \leq \delta, \qquad (22)$$

when $C$ is small enough so that $C(1 + 2 L_\xi \max_i \eta_i) \leq 1$. Therefore, $\|\vec{\xi}_{\tau+1}\|_1 \leq \delta$ and $\Phi(\vec{\theta}_{\tau+1}) \leq \max_{\vec{\theta}' : \|\vec{\xi}(\vec{\theta}')\|_1 \leq \delta} \Phi(\vec{\theta}')$. Finally, because $|\xi_{t,k}^i| \leq C^2 \delta^2 / 4$ for all $t \geq \tau + 1$, by claim D.3, the potential function is decreasing for all $t \geq \tau + 1$, unless $\|\vec{\xi}\|_t \leq C\delta$. Both make the potential function less than $\max_{\vec{\theta}' : \|\vec{\xi}(\vec{\theta}')\|_1 \leq \delta} \Phi(\vec{\theta}')$ which completes the proof. $\qquad \square$

# E PROOF AND DETAIL FOR THEOREM 4.6

*Proof of lemma 4.7.* Define $P_L(x) := \alpha(L)(x - \beta(L))$ which is increasing for all $L$ because $\alpha(L) > 0$. With $P_L$, $f_{\alpha(L),\beta(L)}(x) = \frac{x}{x + (1-x) \exp(P_L(x))}$. Take $x_1(L) = x_1 := 1 - 1/\alpha(L)$, $x_2(L) = x_2 := f_{\alpha(L),\beta(L)}(x_1(L))$, and $x_3(L) = x_3 := f_{\alpha(L),\beta(L)}(x_2(L))$. We will define $x_0(L)$ later. We will omit $L$ and use $x_1$, $x_2$, and $x_3$.

To define $x_0(L)$, we set $y(L) := \beta(L)/2$, and want to show

$$f_{\alpha(L),\beta(L)}(y) > x_1, \qquad (23)$$

which is equivalent to $(\alpha(L) - 1)(1 - y(L)) \exp(P_L(y(L))) < y(L)$. When $L$ is large enough, we have $\beta(L) \in (\frac{2\beta_\infty}{3}, \frac{4\beta_\infty}{3})$. Additionally because $\alpha(L) > 0$, we have $P_L(y(L)) = -\frac{1}{2}\alpha(L)\beta(L) < -\frac{1}{3}\alpha(L)\beta_\infty$. Therefore, we have $(\alpha(L) - 1)(1 - y(L))e^{P_L(y(L))} \leq \alpha(L)e^{-\alpha(L)\frac{\beta_\infty}{3}} < \frac{\beta_\infty}{3} < y(L)$ when $L$ is large enough, and prove eq. (23). On the other hand, because $\beta(L) < \frac{4}{3}\beta_\infty$ and

$P_L(\beta(L)) = 0$, we have $f_{\alpha(L),\beta(L)}(\beta(L)) = \beta(L) < \frac{1}{2}(\beta_\infty + 1)$. Moreover, $\frac{1}{2}(\beta_\infty + 1) < 1 - \frac{1}{\alpha(L)}$, holds when $L$ is large enough. These two imply

$$f_{\alpha(L),\beta(L)}(\beta(L)) = \beta(L) < x_1(L). \tag{24}$$

Combining eqs. (23) and (24), by intermediate value theorem, there exists $x_0$ such that

$$f_{\alpha(L),\beta(L)}(x_0) = x_1 \text{ with } y < x_0 < \beta(L) < x_1. \tag{25}$$

Now we show $x_3 < x_0$. With $1 - x_1(L) = 1/\alpha(L)$ and $0 < x_1 < 1$, we have $x_2 = f_{\alpha(L),\beta(L)}(x_1) = \frac{x_1}{x_1 + \alpha(L)^{-1} \exp(P_L(x_1))} \leq \frac{\alpha(L)}{\exp(P_L(x_1))}$. Because $\beta_\infty < 1/2$, when $L$ is large enough, $x_1(L) = 1 - \frac{1}{\alpha(L)} > \frac{1}{4}(2\beta(L) + 3)$ and, thus, $P_L(x_1) > \alpha(L)\left(\frac{3}{4} - \frac{\beta(L)}{2}\right)$. Therefore, we have

$$x_2 \leq \alpha(L) \exp\left(-\alpha(L)\left(\frac{3}{4} - \frac{\beta(L)}{2}\right)\right) \tag{26}$$

Finally, because $P_L(x_2) \geq P_L(0) = -\beta(L)$, we get

$$\begin{aligned}
x_3 = & f_{\alpha(L),\beta(L)}(x_2) \leq \frac{x_2}{x_2 + (1 - x_2)\exp(-\alpha(L)\beta(L))} \\
= & \frac{x_2 \exp(\alpha(L)\beta(L))}{1 + x_2(\exp(\alpha(L)\beta(L)) - 1)} \\
\leq & x_2 \exp(\alpha(L)\beta(L)) \qquad\qquad \text{(since } \exp(\alpha(L)\beta(L)) > 1 \text{ when } L \text{ is large)} \\
\leq & \alpha(L) \exp\left(\alpha(L)\left(\beta(L) - \left(\frac{3}{4} - \frac{\beta(L)}{2}\right)\right)\right) \qquad\qquad \text{(eq. (26))} \\
= & \alpha(L) \exp\left(\frac{3\alpha(L)}{2}\left(\beta(L) - \frac{1}{2}\right)\right)
\end{aligned}$$

Finally, because $\beta_\infty < 1/2$, $x_3$ converges to 0 when $L$ is large enough. Therefore,

$$x_3 < y < x_0. \tag{27}$$

By eqs. (25) and (27), we have $x_3 < x_0 < x_1$ which completes the proof. $\qquad\square$

