# OpenReview forum: "Multi-agent Performative Prediction: From Global Stability and Optimality to Chaos"
_ICLR.cc/2022/Conference — ICLR 2022 Submitted_

### Official Review · Reviewer_xAh2 · 2021-11-02

**Correctness:** 4
**Technical Novelty And Significance:** 2
**Empirical Novelty And Significance:** 2
**Recommendation:** 6
**Confidence:** 4

**Main Review:**

strengths:

(1) The problems and settings are presented clearly, the results and intuitions are also explained without obscurity. The paper is well written and easy to follow.

(2) The results look correct and good as far as I have checked. As claimed, this is the first time "where such formal chaotic results are established in settings related to performative prediction and supervised learning more generally". I think this paper makes a reasonable contribution.

weakness:

(1) The technical novelty and difficulty seem restricted to me.

(1a) The first main result of "using small learning rate EGD is stable" is kind of a standard result as expected, and also the techniques are based on standard analysis (potential functions and approximating continuous gradient flows).

(1b) The second main result of "using large enough learning rate EGD is Li-Yorke chaotic" is also similar to existing work, e.g., the observation of EGD is chaotic in other games (Palaiopanos et al., 2017). The analysis is based on similar idea of constructing a recurrent behaviour with period three.

As far as I can see, the main difference and novelty is to apply those relevant techniques in performative prediction, which is a bit different with traditional supervised learning. However, performative prediction itself is also a special kind of game / reinforcement learning, which makes the contribution seem not very insightful given we have existing observation of EGD is chaotic in other game settings.

**Summary Of The Paper:**

This paper studied the different behaviours of using exponentiated gradient descent (Def. 2.3) in linear regression with different learning rates. The setting is called performative prediction, which can be viewed as a special case of reinforcement learning (after the model makes a prediction, the environment returns a feedback by changing the data distribution).

The main results consist of two parts, for small enough and large learning rates:

(i) with small enough learning rate, the authors proved that the exponentiated gradient descent is "stable" (asymptotically converges to some global minimizer). The proof is based on standard analysis of studying a potential (surrogate/Lyapunov) function for the mean squared error, and arguing that the gradient flow vanishes only if the potential approaches $0$, which implies that the convergence of the flow. Then the discrete gradient update is approximating the flow well if the stepsize is small enough.

(ii) if the learning rate is larger than some threshold value, the authors then showed that the exponentiated gradient descent became Li-Yorke chaotic, where it can periodically oscillate for infinitely many times. The main idea is to use Eq. (7) to characterize the dynamics, and show that under some conditions Eq. (7) induces a sequence of { x_i }_{ i >= 1}, with some recurrent behaviours as shown in Lemma 4.7.

The authors then verified the theoretical findings using simulations on simple examples.

**Summary Of The Review:**

Overall, I think this paper makes a reasonable contribution, showing that the behaviours of using exponentiated gradient descent differ with using different learning rates. However, given there already exist observations and results about exponentiated gradient descent could be chaotic in several game settings, and the performative prediction studied in this paper is also a special game setting, I consider the contributions and technical novelty in this work on an incremental level.


----update----

I would like to thank the authors for the feedback. My main concerns are addressed and thus I would increase my score.

---

> ### Author Response · Authors · 2021-11-23
> **Response to Reviewer xAh2**
>
> We thank the reviewer for the work, and we have submitted the revised manuscript incorporating your suggestion.
>
> We want to argue that our theoretical contribution is far from trivial and makes some inroads in other well-studied problems. The use of Lyapunov functions, as well as the period three proof, are indeed necessary tools for anyone trying to prove convergence or Li-Yorke chaos in any setting.
>  We have provided a self-contained treatment on multi-agent performative prediction (MAPP), which we believe will be very useful in consolidating a wide range of techniques (performance analysis, Lyapunov theory, chaos) inherently necessary for the study of even simple MAPP settings.
>
> Below we provide more discussion between our paper/techniques and previous works on learning in games.
>
> **Convergent results**
> As mentioned in our result, our convergent result is exact under a small enough but fixed learning rate.  Contrarily, previous works in performative prediction only show approximate convergence where their dynamics converge to a small neighborhood of the performative stable point.  On the other hand, our technique is related to learning in potential games.  Specifically, our dynamics can be seen as a special case of multiplicative weight update (Hedge algorithms) on congestion games.  However, our results are also the first exact convergence result for Hedge algorithms on congestion games.  Kleinberg et al. (2009) only show approximated convergence when the learning rate of Hedge algorithms is small enough.  Palaiopanos et al. (2017) show the linear variant of multiplicative weight update can converge to equilibrium exactly.  Furthermore, we believe our analysis can be extended to exact convergence for the Hedge algorithm with constant step size on general congestion games.
> Besides improvement over previous works, the approximation between discrete gradient update and gradient flow contains novel ideas.  The main challenge is that we do not have a potential function for the discrete process but only have an approximated one, $\Phi$.  To show exact convergence, we show an inductive invariant structure between vanishing components and decrements of $\Phi$ in Claim D3.
>
> **Chaotic results**
> The fact that some of the other systems have been shown to be Li-Yorke chaotic does not imply that this is necessary or even expected that it should behave in such a way in our setting of EGD in performative prediction games.  For example, EGD is provably not Li-Yorke chaotic in zero-sum games as in [1].  Thus, there is no apriori reason to expect either convergence or chaos for EGD in a family of games before one performs a detailed analysis.  Moreover, as we point out in our paper, ever since the important first result of Li-Yorke chaos in games by Palaiopanos et al., there has been a strong interest in understanding the emergence of chaos in several different games, Machine Learning settings [2,3,4,5]. We believe that our paper adds an exciting chapter to this growing literature.
>
> **Optimality**
> Finally, there is another critical conceptual difference between our setting and that of general equilibrating game dynamics. In our setting, we do not just prove convergence in the case of small steps sizes, but we furthermore prove that we convergence to an optimal state even though our dynamics in Eqn. (3) may have multiple fixed points. This once again requires arguments that are totally orthogonal to both Lyapunov theory, and this is, for example, a question not explored in Palaiopanos et al. (2017).  Thus importantly, we do not just do stability vs. chaos analysis alone but also combine this with performance analysis. The only other paper that we know that explores this interplay is [2] but
> their optimality/convergence analysis is done only on small games with just two strategies, whereas our results and techniques scale up to arbitrarily large, complex games.
>
> For all the above reasons, we believe that we offer significant technical insights into this important emergent field.
>
> [1] Bailey, J.P. and Piliouras, G., 2018, June. Multiplicative weights update in zero-sum games. In Proceedings of the 2018 ACM Conference on Economics and Computation (pp. 321-338).
>
> [2] Chotibut, Thiparat, et al. "The route to chaos in routing games: When is Price of Anarchy too optimistic?." NeurIPS 20
>
> [3] Bielawski, Jakub, et al. "Follow-the-Regularized-Leader Routes to Chaos in Routing Games." ICML 21
>
> [4] Cheung, Yun Kuen, et al. "Learning in Markets: Greed Leads to Chaos but Following the Price is Right." IJCAI 21
>
> [5] Cheung Yun Kuen and Tao, Yixin. Chaos of Learning Beyond Zero-sum and Coordination via Game Decompositions. ICLR 21.

---

### Official Review · Reviewer_jNt2 · 2021-11-02

**Correctness:** 3
**Technical Novelty And Significance:** 3
**Empirical Novelty And Significance:** 2
**Recommendation:** 6
**Confidence:** 3

**Main Review:**

Strengths:
The framework of performative predictions is very interesting, with potential applications that are beyond the standard supervised learning methods. Its extension to a multiagent setting widens the  applicability of the framework in the real world.
The paper studies the emergence of instabilities both theoretically and empirically.

Weaknesses:
1. I am not convinced by some of the derivations:
1.1. Proof of Prop 3.1 should be very simple but it is confusing. The authors refer to the "KKT condition" --which is itself confusing as the KKT are a set of conditions that have to hold together, not just one--, but it is not clear which one of the KKT conditions is referred by the inequality on the partial derivatives. What does it mean that the partial derivative on one component, $k$, is less or equal than the partial derivative in another component, $l$?
1.2. In the second part of the proof of Prop 3.1, the authors use the positive definite Hessian to prove strict convexity, do they mean strong convexity?
1.3. A performative optimal point (Def 2.1) is just a Pareto solution of the multiobjective problem (indeed it could probably be generalised to any convex combination of each agent's losses). On the other hand, performative stability is a global solution of the objective problem, and is a much stronger condition than the Performative (Pareto) optimality. Indeed, it is trivial to show that the former implies the latter, so I don't understand the comment before Prop. 3.2.
1.4. In Eq. (4), the authors define the fixed point of (3) as the average gradient. Why is so? The argument at the end of page 5, in which they multiply numerator and denominator by the average gradient seems rather loose, since you could multiply both factors by any other value and still claim that this value is the convergence point. I think they have to formally prove that:
$\lim_{k \rightarroy \infty} g_k^i(\theta) = \overline{g}^i(\theta) \in \Re^d$.
Perhaps this apparently circular reasoning can be easily fixed by proving Theorem 4.1 before Eq. (4).

2. Discussion of connection with related approaches is missing.
2.1. The proposed approach reminds me of the replicator dynamics from evolutionary game theory, for which the emergence of chaos has also been studied.
2.2. In order to prove performative stability, could the authors rely on the strong-monotonicity of the gradient and well known results for finite-dimensional variational inequality theory (see, e.g., Scutari et al., 2010)?
2.3. Conditions on the learning rate for the chaotic behaviour of gradient descent have been studied in the literature. How the current results relate to these previous results?

3. The simulations show oscillatory behaviour, but it is not clear from the experiments that this is actually chaotic behaviour. I would have expected to see at least sensitivity to initial conditions, and ideally negative Lyapunov exponents computed from the time series.


Minor comment:
It is clear that the algorithm will converge for small enough learning rate, and it can oscillate or diverge beyond that threshold. Some motivation on why it is important to study the chaotic behaviour in particular would help to appreciate the impact of this work.


(Scutari et al., 2010), Gesualdo Scutari, Daniel P. Palomar, Francisco Facchinei, and Jong-Shi Pang, "Convex Optimization, Game Theory, and Variational Inequality Theory", IEEE Signal Processing Magazing 35, May 2010


**Summary Of The Paper:**

The paper introduces a multiagent extension of the performative prediction framework in which multiple agents try to predict the same outcome, which influences the outcome they want to predict. The paper then shows scenarios and conditions for stability and for chaotic behaviour.


**Summary Of The Review:**

The extension of the performative predictions framework to multiagent systems is an interesting problem. and even the scenario under study with linear predictors and MSE loss greatly simplifies the analysis, I appreciate this is a first step.
This is mainly a theoretical paper, but some of the mathematical derivations have raised some questions.

Metareview
----------------
The authors have responded to most of my comments clearing my concerns on the derivations. I think that an experiment showing sensitivity to initial conditions is missing. Although the novelty seems incremental with respect to previous works, there are a few small innovations, so I am increasing my score to 6.

---

> ### Author Response · Authors · 2021-11-23
> **Response to Reviewer jNt2 (1/3)**
>
>
> We thank the Reviewer for their work. We address their points below:
>
> **2. Strongly or strictly convex**
> Strict convexity is sufficient to show the uniqueness and existence of minimum value, because the space is compact and $\Phi$ is continuous.
>
> **3. Definition of performative optimal point**
>
> The performative optimality does not implies the performative stable point.  For performative optimal point, the variable $\vec{\boldsymbol{\theta}}$ affects not only the first argument but also the second one,
> $$\arg\min_{\vec{\boldsymbol{\theta}} = (\boldsymbol{\theta}^1, \dots, \boldsymbol{\theta}^n)}\sum_i \ell(\vec{\boldsymbol{\theta}}, \boldsymbol{\theta}^i)$$
> On the other hand, for performative stable point $\vec{\boldsymbol{\theta}}^*$, the variable only affects the second argument,
> $$\arg\min_{\boldsymbol{\theta}^i} \ell(\vec{\boldsymbol{\theta}}^*, \boldsymbol{\theta}^i)$$
>
> A preformative optimal point $\vec{\boldsymbol{\theta}}^*$ ensures the total loss $\sum_i \ell(\vec{\boldsymbol{\theta}}^*, (\boldsymbol{\theta}^*)^i)$  is minimized.  However, one player may deviate and change his predictive model from $(\boldsymbol{\theta}^*)^i$ to $\boldsymbol{\theta}^i$ and decrease his decoupled performative loss, $\ell(\vec{\boldsymbol{\theta}}^*, \boldsymbol{\theta}^i)<\ell(\vec{\boldsymbol{\theta}}^*, (\boldsymbol{\theta}^*)^i)$.
>
> **4. Fixed points of mapping (3)**
> If $\vec{\boldsymbol{\theta}}^*$ is a fixed point of the mapping (3), we have
> $$(\vec{\boldsymbol{\theta}}^*)^i_{k} = \frac{(\vec{\boldsymbol{\theta}}^*)^i_{k}\exp(-\eta_i g^i_k(\vec{\boldsymbol{\theta}}^*))}{\sum_{l = 1}^d (\vec{\boldsymbol{\theta}}^*)^i_{l}\exp(-\eta_i g^i_l(\vec{\boldsymbol{\theta}}^*))}\text{ for all } i\in [n], \text{ and } k\in [d].$$
> Thus, $\vec{\boldsymbol{\theta}}^*$ is a fixed point if and only if for all $i\in [n]$, $k, k'\in [d]$ with $(\theta^*)^i_k, (\theta^*)^i_{k'}>0$,
> $g_k^i(\vec{\boldsymbol{\theta}}^*) = g_{k'}^i(\vec{\boldsymbol{\theta}}^*)$ which is equivalent to equation (4), $g^i_k(\vec{\boldsymbol{\theta}}^*) = \bar{g}^i(\vec{\boldsymbol{\theta}}^*)$ for all $i\in[n]$ and $k\in[d]$ with $(\theta^*)^i_k>0$.
> In our revised version, we also emphasize that we use subscript $l$ and $k\in[d]$ to denote the coordinates of the feature, and $t\ge 0$ to denote the time.

---

> > ### Author Response · Authors · 2021-11-23
> > **Response to Reviewer jNt2 (2/3)**
> >
> > **1. KKT conditions**
> > The collection of inequalities on the partial derivatives is equivalent to the collection of KKT conditions for each player's minimization problem of decoupled performative loss.  We believe the derivation is fairly standard.  See below for more details.
> >
> > Given a collection of predictive model $\vec{\boldsymbol{\theta}}'$, agent $i$ wants to maximize its decoupled performative loss
> > $L_i(\boldsymbol{\theta}^i) := \ell(\vec{\boldsymbol{\theta}}', \boldsymbol{\theta}^i)$ which is convex.  Formally,
> > $$\begin{aligned}
> > \min_{\boldsymbol{x}} \quad & L_i(\boldsymbol{x})\\
> > \textrm{s.t.} \quad & x_k\ge 0, \forall k\in [d]\\
> >   &\sum_{k\in [d]} x_k = 1\\
> > \end{aligned}$$
> > By the KKT conditions for the above convex problems, the primal $\boldsymbol{x}^*$, and dual variables $\sigma_1, \dots, \sigma_d$ and $\gamma$ satisfy
> > $$\begin{aligned}
> >     &x^*_k\ge 0, k\in[d]\\\\
> >     &\sum_k x^*_k = 1\\\\
> >     &\sigma_k\ge 0, k\in [d]\\\\
> >     &\sigma_k x^*_k = 0, k\in [d]\\\\
> >     &\frac{\partial}{\partial x_k}L_i(\boldsymbol{x}^*)-\sigma_k+\gamma = 0, k\in [d].
> > \end{aligned}$$
> > For all $l\in [d]$, by the fifth equation, $\frac{\partial}{\partial x_l}L_i(\boldsymbol{x}^*) = \sigma_l-\gamma$ which is no less than $-\gamma$ by the third equation.  Additional if $x^*_k>0$, $\sigma_k = 0$ by the fourth equation, and $\frac{\partial}{\partial x_k}L_i(\boldsymbol{x}^*) = -\gamma$.  Therefore, for all $k,l\in [d]$ with $x^*_k>0$,
> > $$\frac{\partial}{\partial x_k}L_i(\boldsymbol{x}^*) \le \frac{\partial}{\partial x_l}L_i(\boldsymbol{x}^*).$$
> > We have shown that the optimality of $\boldsymbol{x}^*$ implies the above equation.  Conversely, if $\boldsymbol{x}^*\in \Theta$ satisfies the above equation, the KKT conditions also hold.  Therefore the above equation is an sufficient and necessary condition for maximizing agent $i$' decoupled performative loss.
> >
> > By definition of $L_i$, we have $\frac{\partial}{\partial x_k}L_i(\boldsymbol{x}^*) = \frac{\partial}{\partial x_k}\ell(\vec{\boldsymbol{\theta}}', \boldsymbol{x})|_{\boldsymbol{x} = \boldsymbol{x}^*}$.
> >
> > Therefore, $\vec{\boldsymbol{\theta}}^*$ is performative stable if and only if for all agent $i$, $\frac{\partial}{\partial x_k}\ell(\vec{\boldsymbol{\theta}}^*, \boldsymbol{x})\mid_{\boldsymbol{x} = (\vec{\boldsymbol{\theta}}^*)^i}$ which is less than $\frac{\partial}{\partial x_l}\ell(\vec{\boldsymbol{\theta}}^*, \boldsymbol{x})|_{\boldsymbol{x} = (\vec{\boldsymbol{\theta}}^*)^i}$ for all $k, l\in [d]$ with $\theta^i_k>0$.  That is
> > $g^i_k(\vec{\boldsymbol{\theta}}^*)\le g^i_l(\vec{\boldsymbol{\theta}}^*)$
> >  for all $i\in [n]$ and $k, l\in [d]$ with $\theta^i_k>0$.

---

> > > ### Author Response · Authors · 2021-11-23
> > > **Response to Reviewer jNt2 (3/3)**
> > >
> > >
> > > **7. Replicator dynamics** Chaos under replicator dynamics is not possible in our setting due to the existence of a Lyapunov/potential function. This is actually a problem with replicator in these settings. Their smoothness can actually hide instability results that arise in their real world, discrete-time implementation.
> > >
> > > **8. strongly monotone games** Our techniques do not leverage ideas from strongly monotone games.
> > >
> > > **9. chaotic gradient descent** We are familiar with the following chaotic results in the literature for GD (as part of analysing FTRL with different regularizers)[1-3]. These works show a different type of chaos via a volume increase analysis. This does not imply Li-Yorke chaos. Actually some of these dynamics are provably not Li-Yorke chaotic as they diverge away from equilibria thus not allowing the possibility of any periodic orbits.
> > >
> > > [1] Yun Kuen Cheung, Georgios Piliouras.
> > > Vortices Instead of Equilibria in MinMax Optimization: Chaos and Butterfly Effects of Online Learning in Zero-Sum Games. COLT 2019: 807-834.
> > >
> > > [2] Yun Kuen Cheung, Georgios Piliouras.
> > > Chaos, Extremism and Optimism: Volume Analysis of Learning in Games. NeurIPS 2020.
> > >
> > > [3] Yun Kuen Cheung, Yixin Tao.
> > > Chaos of Learning Beyond Zero-sum and Coordination via Game Decompositions. ICLR 2021.
> > >
> > >
> > > **10. Lyapunov exponents** Positive (not negative) Lyapunov exponents is a typical heuristic way for arguing about instability of trajectories after infinitesimal perturbations. In our case such heuristic arguments are not necessary as Li-Yorke chaos provably implies positive topological entropy (in fact any odd period orbit suffices) [1]. This means that there is an exponential growth of the number of distinguishable orbits and thus arbitrarily small perturbations can lead quickly to macroscopically large effects (what is typically referred to informally as the ``butterfly effect").
> > >
> > > [1] Michal Misiurewicz. 1979. Horseshoes for mapping of the interval. Bull. Acad. Polon.
> > > Sci. S\'{e}r. Sci. 27 (1979), 167–169.

---

### Official Review · Reviewer_RUeS · 2021-11-06

**Correctness:** 4
**Technical Novelty And Significance:** 2
**Empirical Novelty And Significance:** 2
**Recommendation:** 6
**Confidence:** 3

**Main Review:**

# Main Review

The paper is clearly written and its main theorems are supported by proofs and numerical simulations.
The strength of the paper is that it extends
a relatively new model (2020) to a cooperative multi-agent setting.
The extension to the multi-agent case has both positive
and negative results
(convergence and chaos),
and the authors provide theoretical justification for both occurances by leveraging
past work on learning in potential games.
These results are already known for other classes of conjestion games, so the authors support the
previous literature
by showing that they are also true for constrained linear models in this class of problems.

We agree with the authors that this work only scratches the surface of the in multi-agent performative prediction.
The model is an original and novel contribution (to the best of my knowledge), but the results are confirmatory. The results are mostly shown by previous work already since the model trivially has a potential function. So it is unclear to me if the work is significant for this venue.
I also think some commentary on stability concept (the second part of definition 2.1) would be helpful towards motivating the significance of the work.
How does the multi-agent stability concept differ from the stability concept in (Perdomo 2020)?
Can you point to the differences (existence, uniqueness or stability..)
or what makes this class much more challenging to solve? This could be useful for future work in understanding when chaos occurs, for example.

The quality of the paper can be improved in some minor ways, especially by paying attention to details.
For example, the last equation on page 3, the pipe operator in the lefthand side is undefined.
Also, the use of \theta_0 is different in section 2.1 and 2.2, and I think it can be confusing.
Section 2.3 just consists of a definition and nothing else. Can you explain it or at the very least provide a reference?
There is a typo in the last sentence of section 3.
Pay attention to capitalization, and espeically in the reference list, where many typos can be found.

(A minor comment on style:
there is a missed opportunity to
maintain the parallelism
of "convergence to chaos" in your title
in Figure 1 by
sorting the figures in that order, ie putting (a) and (d) in the rightmost column.)


**Summary Of The Paper:**

The paper analyzes the performative prediction setting (Perdomo 2020)
where multiple agents perform gradient descent to converge to a performatively optimal point.
The agent are modeled by constrained linear predictive models which are used for linear regression.

The authors show that the learning dynamics
converges to the multi-agent performatively stable point
for small learning rates.
The stable point coincides with the optimal point where the sum of the agent's losses are minimized.
The requirement for convergence is that the hessian of the loss must be positive definite, which is satisfied by existence of a potential function.

The authors also show
that the dynamics exhibits Li-York chaos for large enough learning rates.

**Summary Of The Review:**



Overall the paper has a central point by presenting a framework of multi-agent learning. The model is cooperative, so a potential function exists and learning converges for slow learning rate regime.
A discussion section is lacking, in particular with regard to the significance of the work and results.
What insights can be provided by the analysis from the paper?
Currently the conclusion section include mostly a list of future applications.

The claims in the paper are supported and correct to the best of my knowledge. The significance of the results is debatable but the model is indeed novel. The empirical data shows an example of chaos and two examples of convergence, with and without noise. In my opinion, the model is novel enough for me to recommend a 6, but the technical results are expected and perhaps maybe that can push it to below the acceptance threshold.

---

> ### Author Response · Authors · 2021-11-23
> **Response to Reviewer RUeS**
>
> We thank the reviewer for the work and their support for our paper.  We have submitted the revised manuscript which incorporated your suggestion.
>
> The use of Lyapunov functions, as well as the period three proof, are indeed necessary tools for anyone trying to prove convergence or Li-Yorke chaos in any setting. Although of course there are similarities to prior work, there are also significant technical differences, and we believe we make some new inroads in well-studied problems.
> We provide more discussion in our response to reviewer xAh2.
>
> The notion of multi-agent performative stability generalizes the original performative stability. It coincides with the performative stability when the number of players is one.
>
> We agree with the reviewer that our discussion session was lacking. We have edited the conclusion section to consolidate some high-level points in the revision. We believe an important and largely underexplored aspect of ML is what happens when ML systems form close-loop as they both affect their environment and thus affect each other. Our simple and we believe natural setting shows that these systems have some implicit carrying capacities, and once enough algorithmic power is introduced either via a few large agents or very many small once instability can be inevitable and lead to performance degradation. This phenomenon cannot be understood by studying ML systems in isolated sandboxes and moreover sets up new theoretical as well as practical challenges. We believe that our paper sets the stage for exciting and important follow-up work. The notions of performance, Lyapunov theory, and chaos theory will be part of their vocabulary.
>
> Thank you for the nice catch with Fig 1. That is a nice idea! We implemented it. Thank you for all your thoughtful feedback.

---

### Decision · Program_Chairs · 2022-01-20

**Decision:**

Reject

**Comment:**

In this paper, the authors extend the performative prediction framework of Perdomo et al. (2020) to a multi-agent, game-theoretic setting, and they examine how and when multi-agent performative learning may lead to performative stability/optimality.

The authors' results and contributions can be summarized as follows:
- They consider a multi-agent location-scale distribution map with parameters constrained in a simplex, and they study the dynamics of an exponentiated gradient descent algorithm (EGDA for short) inspired by Kivinen and Warmuth (1997).
- If the learning rate is small enough, the authors show that EGDA converges to a performatively stable point (under the same assumptions that guarantee existence of a convex potential).
- On the other hand, if the learning rate is large, the algorithm behaves chaotically.

The reviewers' initial assessment was mixed, but after the authors' rebuttal, some concerns were partially addressed and the scores of the paper were upgraded to borderline positive. On a point-by-point basis, the authors' result on the convergence of EGDA with a small learning rate was appreciated by the reviewers, but it was not otherwise deemed significant enough relative to existing convergence results for gradient methods. Instead, most of the discussion centered on the authors' result on chaos (Theorem 4.6), which was viewed as the most significant contribution of the paper. However, continued discussion between committee members revealed that this result follows directly from Theorem 3.11 and Corollary 3.12 of the arxiv preprint "A family of chaotic maps from game theory" by T. Chotibut, F. Falniowski, M. Misiurewicz, and G. Piliouras <https://arxiv.org/abs/1807.06831>, which is not discussed in the paper. [As was pointed out, the update map (7) of the paper coincides with the update rule (7) of the arxiv preprint, and the proof techniques are likewise identical.]

This overlap with previous work was considered a "big omission" and it pushed the paper below the acceptance threshold. In the end, the paper was not supported by any of the reviewers, so a "reject" recommendation was made.